# TEMPERATURE OPTIMIZATION FOR BAYESIAN DEEP LEARNING

## ABSTRACT

The Cold Posterior Effect (CPE) is a phenomenon in Bayesian Deep Learning (BDL), where tempering the posterior to a cold temperature often improves the predictive performance of the posterior predictive distribution (PPD). Although the term 'CPE' suggests colder temperatures are inherently better, the BDL community increasingly recognizes that this is not always the case. Despite this, there remains no systematic method for finding the optimal temperature beyond grid search. In this work, we propose a data-driven approach to select the temperature that maximizes test log-predictive density, treating the temperature as a model parameter and estimating it directly from the data. We empirically demonstrate that our method performs comparably to grid search, at a fraction of the cost, across both regression and classification tasks. Finally, we highlight the differing perspectives on CPE between the BDL and Generalized Bayes communities: while the former primarily emphasizes the predictive performance of the PPD, the latter prioritizes the utility of the posterior under model misspecification; these distinct objectives lead to different temperature preferences.

## 1 INTRODUCTION

The last two decades have seen substantial advances in the development of machine learning tools. This has led to deployment across a range of application domains, from image recognition and natural language processing to autonomous vehicles and medical diagnostics, to name a few. Driven in no small part by the technological advances that have enabled large-scale training of high-fidelity models, these models obtain impressive generalization performance and accuracy that are not readily explained with classical statistical wisdom.

One promising avenue that has been explored for explaining and enhancing performance, as well as providing much-needed robustness guarantees, is **Bayesian deep learning** (BDL). However, extending classical Bayesian techniques to the machine learning setting is a formidable task—the scale of both the data and model size render naïve extensions intractable. Combined with the fact that neural network models are typically singular (Wei et al., 2022), the mismatch between the size of the data and the number of parameters makes many of the classical asymptotics inappropriate when applied to problems of interest. Furthermore, a number of phenomena have been observed when analyzing deep learning models that were previously considered curiosities or corner-cases in the classical setting.

A prominent example of this is the so-called **cold posterior effect** (CPE). Named in analogy with statistical physics, tempered posteriors are obtained by raising the posterior density to the power of an artificial inverse-temperature parameter. The CPE refers to improved generalization performance of the **posterior predictive density** (PPD) in both regression (Adlam et al., 2020) and classification (Wenzel et al., 2020) tasks when the temperature $T$ is taken to be cold with $0 < T < 1$. This peculiar effect is frequently viewed as a hack to improve generalization performance, and has led to a plethora of works that attempt to explain or 'fix' the CPE, see Section 5 for a literature review.

**Contribution.** There is growing recognition in the BDL community that, despite the term 'CPE', colder temperatures do not always result in better predictive performance for the PPD (Adlam et al., 2020; Zhang et al., 2024). Unfortunately, the common approach of temperature tuning via grid search is computationally expensive, as it requires extra posterior sampling for multiple $T$ across

the grid[1]. Sampling from a cold posterior is challenging and requires careful tuning of the sampler with temperature diagnostics (Wenzel et al., 2020). To address these issues, we propose a data-driven method to select an appropriate temperature. To the best of our knowledge, no such dedicated tool exists without appealing to intermediate approximations (e.g., variational inference (Laves et al., 2021)). Our method only requires maximizing a likelihood function to find a suitable temperature. This can usually be done as part of the sampler warm-up phase, and importantly, does not require any extra posterior sampling.

## 2 TEMPERED POSTERIORS AND THE COLD POSTERIOR EFFECT

Let $\{q(x, y), p(x, y|\theta), p(\theta)\}$ form a triplet representing the truth-model-prior, where the data-generating mechanism is $q(x, y) = q(y|x)q(x)$, the model $p(x, y|\theta) = p(y|x, \theta)q(x)$ is indexed by $\theta \in \Theta \subseteq \mathbb{R}^d$ representing neural network weights, and the prior on $\theta$ is $p(\theta)$. Here, we focus on a supervised learning setup with a training dataset $\mathcal{D} = \{(x_i, y_i)\}_{i=1}^n$ containing $n$ observations drawn from $q(x, y)$. The standard Bayesian update is derived from Bayes' theorem, resulting in the *standard posterior* distribution,

$$p(\theta|\mathcal{D}) \propto p(\theta) \prod_{(x,y)\in\mathcal{D}} p(y|x, \theta). \tag{1}$$

By introducing a 'temperature' parameter, we obtain a family of *tempered posteriors* $p_\beta(\theta|\mathcal{D})$ by raising the likelihood and prior to the power of the *inverse temperature* $\beta := \frac{1}{T}$ for $\beta > 0$,

$$p_\beta(\theta|\mathcal{D}) \propto p(\theta)^\beta \prod_{(x,y)\in\mathcal{D}} p(y|x, \theta)^\beta. \tag{2}$$

The CPE describes a phenomenon in Bayesian deep learning where the PPD,

$$p_\beta(y|x, \mathcal{D}) := \mathbb{E}_{p_\beta(\theta|\mathcal{D})}[p(y|x, \theta)] = \int p(y|x, \theta)p_\beta(\theta|\mathcal{D})\, d\theta, \tag{3}$$

which is constructed from Bayesian model averaging, can achieve better performance (in terms of the test LPD defined below) in regression and classification by artificially tempering the posterior to $\beta > 1$ (Wenzel et al., 2020). While our construction in (2) follows the convention of tempering both the likelihood *and* the prior (Wenzel et al., 2020; Fortuin et al., 2022), improvements in performance have also been observed when only tempering the likelihood (Aitchison, 2021; Bachmann et al., 2022; Kapoor et al., 2022).

Define the test log predictive density (LPD) of a predictive density $\hat{p}(y|x)$ as

$$\text{LPD}(\hat{p}(y|x)) := \mathbb{E}_{q(x,y)} \log \hat{p}(y|x),$$

where the hat over $p$ indicates dependence on the training data $\mathcal{D}$, and the expectation is only taken over the 'new' observation $(x, y)$. Note that the test LPD is sometimes referred to as the 'test log-likelihood' in the literature, and the negative test LPD is often called 'negative log-likelihood (NLL)'. However, we will avoid these terms as they can sound ambiguous. It can be shown that a predictive density $\hat{p}(y|x)$ with a higher test LPD is closer to the truth $q(y|x)$ in the sense that the Kullback-Leibler (KL) divergence $D_{\text{KL}}(q(y|x)\|\hat{p}(y|x))$ is smaller. Our primary interest in this work is to select $\beta$ such that the test LPD of (3) is high.

### 2.1 OUTLINE

Following the reasoning that the aleatoric uncertainty in the data can be quantified by $\beta$ (Adlam et al., 2020; Kapoor et al., 2022), we advocate a likelihood-based approach to select $\beta$ directly from the data; details are presented in Section 3. We empirically show that our method can select a $\beta$ that is near-optimal according to the LPD metric. Experimental results are presented in Section 4.

We then conclude with a detailed discussion aimed at dispelling some common misconceptions surrounding the CPE. In particular, we review recent work in the BDL community that explains

---

[1]Consider ResNet20-CIFAR10 for example. Our approach requires only 35 minutes to find the optimal $T$, and an additional 327 minutes to compute the posterior at that optimal $T$. In contrast, a grid search across 9 temperatures takes a substantial 2944 minutes, resulting in a 8x speedup with our method.

why colder temperatures do not always lead to better performance in terms of the test LPD metric. We also address the converse hypothesis sometimes encountered outside the BDL literature—that warmer temperatures are often better. This hypothesis may hold in certain contexts, such as ensuring posterior consistency (Grünwald, 2012), but it generally does not apply to improving test LPD. Specifically, in Sections 5 and 6, we examine existing literature on CPE from the perspectives of the BDL and Generalized Bayes (GB) communities, respectively. We emphasize that these two communities prioritize different objectives: in BDL, the primary focus is on maximizing the predictive performance of the PPD, while in GB, the emphasis lies on the utility of the posterior under model misspecification. Furthermore, the types of statistical models and datasets explored in each community are notably distinct. In BDL, large-scale neural networks are trained on extensive, nearly noiseless datasets, whereas in GB, the datasets are typically smaller, noisier, and the models are classical regular statistical models. Consequently, it is not surprising that each community arrives at different recommendations regarding temperature tuning.

## 3 TEMPERATURE SELECTION FOR TEST LPD

It is commonly believed that the PPD (3) at $\beta = 1$ is optimal for test LPD when the model is well-specified (Adlam et al., 2020; Aitchison, 2021). However, this is not always the case (see Appendix A for a counter-example). In fact, Zhang et al. (2024) argues that in the likelihood tempering case, $\beta = 1$ can only be optimal if the expected Gibbs training loss remains unchanged with the inclusion of new data. We follow the reasoning in Zhang et al. (2024) and reach a similar conclusion for posterior tempering; see Appendix B for details. This motivates the development of an efficient method to select the optimal $\beta$.

If we could have some theoretical grasp on how temperature affects the test LPD of the PPD in (3), that is, $\text{LPD}(p_\beta(y|x, \mathcal{D}))$, then it might suggest a methodology for temperature selection. A little known result to the BDL and GB communities is the following from singular learning theory (Watanabe, 2010a, Lemma 3), which applies to both regular and singular models $p(y|x, \theta)$:

$$-\mathbb{E}_{\mathcal{D}} \text{LPD}(p_\beta(y|x, \mathcal{D})) = -\mathbb{E}_{\mathcal{D}} \mathbb{E}_{q(x,y)} \log p_\beta(y|x, \mathcal{D})$$

$$= -\mathbb{E}_{q(x,y)} \log p(y|x, \theta_\dagger) + \left[ \frac{\lambda - \nu(\beta)}{\beta} + \nu(\beta) \right] \frac{1}{n} + o\left(\frac{1}{n}\right) \quad (4)$$

where $\theta_\dagger \in \arg\min_\theta D_{\text{KL}}(q(y|x)\|p(y|x, \theta))$. Here $\lambda$ and $\nu$ are strictly positive numbers, respectively known as the learning coefficient and singular fluctuation. They are invariants of the underlying truth-model-prior triplet. Note that $\lambda$ is independent of $\beta$ while $\nu$ is a (complex) function of $\beta$. The functional dependence of $\nu$ on $\beta$ is unknown in the current singular learning theory literature.

Some comments on (4) are in order. This result allows for both misspecification ($\theta_\dagger$ is not necessarily such that the KL divergence between the truth and $p(y|x, \theta_\dagger)$ is zero) and singular models singular models $p(y|x, \theta)$, including neural networks, as well as classical models satisfying standard regularity conditions. On the other hand, it is an asymptotic result in the sample size $n$, and hence the prior plays no role. We also remark that the relation in (4) pertains to the *average* (negative) test LPD of the PPD in (3), with the average taken over the training set $\mathcal{D}$, as indicated by the notation $\mathbb{E}_{\mathcal{D}}$. More precise interpretation of (4) can be divided into two settings:

**Regular models.** For well-specified regular models, $\lambda = \nu(\beta) = d/2$ for all $\beta$ (Watanabe, 2009), and the second term, $\left[ \frac{\lambda - \nu(\beta)}{\beta} + \nu(\beta) \right] \frac{1}{n}$, reduces to $\frac{d}{2n}$. This implies that for large $n$, temperature has little impact (though it may appear in higher-order terms in the expansion). This aligns with recent work by McLatchie et al. (2024), which arrives at a similar conclusion using different techniques. Under model misspecification, the second term can alternatively be expressed as $\frac{\beta}{n}\mathbb{E}_{\mathcal{D}}V(n)$ (Watanabe, 2010b), where $V(n)$ represents the functional variance and now depends on the temperature in a non-trivial manner.

**Singular models.** The asymptotic relation in (4) also applies to singular models, e.g., neural network models (Wei et al., 2022). However, the theoretical values of $\lambda$ are generally unknown, except for a few simple models, such as one-layer tanh or reduced rank regression, in the well-specification setting (Yamazaki & Watanabe, 2003; Aoyagi & Watanabe, 2005; Rusakov & Geiger,

2005; Zwiernik, 2011). Similarly, the singular fluctuation and its temperature dependence are unknown, even in the well-specification setting.

Since our setup involves modern deep learning models, we are dealing with singular models, where $\lambda$ and $\nu$ are unknown. While methods to estimate $\lambda$ and $\nu$ from training data do exist (Lau et al., 2023; Watanabe, 2010a), they require posterior sampling over neural network weights. In principle, we could use these sample-based estimates to select the optimal $\beta$, but this would require posterior sampling at multiple temperatures and is no better than a grid search. As this approach is computationally challenging for deep learning applications, we are motivated to propose a data-driven technique for efficiently determining the optimal temperature, which we detail in the next section.

### 3.1 Selecting Temperature using the Tempered Model

In this section, we introduce a method to select $\beta$ that corresponds to high test LPD in both regression and classification tasks. Our approach is grounded in the insight that the tempered posterior can be reframed as the posterior from an alternative model-prior pair (Zeno et al., 2020; Zhang et al., 2024). Following these works, we define the *tempered model*,

$$p(y|x, \theta, \beta) := \frac{p(y|x, \theta)^\beta}{\int p(y'|x, \theta)^\beta \mathrm{d}y'}. \tag{5}$$

From here, the tempered posterior in (2) can be equivalently expressed as

$$p_\beta(\theta|\mathcal{D}) \propto \tilde{p}(\theta|\beta) \prod_{(x,y)\in\mathcal{D}} p(y|x, \theta, \beta), \tag{6}$$

where the 'rest of the terms', $\tilde{p}(\theta|\beta) \propto p(\theta)^\beta \prod_{x\in\mathcal{D}} \int p(y'|x, \theta)^\beta \mathrm{d}y'$, can be viewed as a prior on $\theta$ with the 'normalizing constant' being a function of $x$ in $\mathcal{D}$. Note that $\tilde{p}(\theta|\beta)$ can be seen as an input-dependent prior (Zeno et al., 2020) but we suppress the dependence on $x$ in the notation. The (inverse) temperature $\beta$ has an intuitive interpretation of controlling the 'spikiness' of the tempered model. We give two examples with Gaussian (regression) and categorical (classification) models:

**Regression.** Given an arbitrary scalar function $\mu(x; \theta)$ and a fixed, known variance $\sigma^2$, we have a Gaussian model: $p(y|x, \theta) = \mathcal{N}(y|\mu(x; \theta), \sigma^2)$. This leads to

$$p(y|x, \theta, \beta) = \mathcal{N}\left(y\,\middle|\,\mu(x; \theta), \frac{\sigma^2}{\beta}\right), \quad \tilde{p}(\theta|\beta) \propto p(\theta)^\beta \left(\frac{2\pi\sigma^2}{\beta(2\pi\sigma^2)^\beta}\right)^{n/2}.$$

Therefore, the temperature effectively scales the model and prior variance. For brevity, we suppress the dependency of $p(y|x, \theta, \beta)$ on the fixed $\sigma^2$ in the notation.

**Classification.** In $K$-class classification, we have $p(y|x, \theta) = f_y(x; \theta)$ for $y \in \{1, \ldots, K\}$, where $f_y(\cdot)$ denotes the $y$-th entry of a softmax output $f$. This leads to

$$p(y|x, \theta, \beta) = \frac{\exp(\beta f_y(x; \theta))}{\sum_{k=1}^K \exp(\beta f_k(x; \theta))}, \quad \tilde{p}(\theta|\beta) \propto p(\theta)^\beta \prod_{x\in\mathcal{D}} \sum_k [f_k(x; \theta)]^\beta.$$

This tempered model is also known as the *tempered softmax* (Hinton et al., 2015; Agarwala et al., 2023). For large $\beta$, the tempered model will concentrate most of the mass in one class, and the converse will encourage a more uniform distribution of mass across all classes. For $\beta > 1$, the prior $\tilde{p}(\theta|\beta)$ will also favor $f$ that concentrates mass in one class.

As $\beta$ can be used to capture aleatoric uncertainty in the data (Adlam et al., 2020; Kapoor et al., 2022), we propose selecting $\beta$ using a maximum likelihood estimator for the tempered model in (5):

$$\hat{\theta}^*, \hat{\beta}^* := \arg\max_{\theta, \beta} \frac{1}{n} \sum_{(x,y)\in\mathcal{D}} [\log p(y|x, \theta, \beta)]. \tag{7}$$

Standard consistency results imply that, under mild conditions, provided the model is regular and *some* tempered model is well-specified, choosing parameters according to (7) recovers the optimal model as $n \to \infty$ (van der Vaart, 1998, Chapter 5.2). In practice, we optimize this using SGD,

and stop when the log-likelihood shows no further improvement on a validation set. To ensure that $\beta$ remains positive, we reparameterize it as $\exp(\log \beta)$ and optimize with respect to $\log \beta$. Further implementation details are provided in Appendix C. We also discussed several variants of our proposed method in Appendix D.

A similar temperature optimization approach was proposed in Guo et al. (2017) for computing a well-calibrated 'plug-in' predictive density $p(y|x, \theta^*_{\text{SGD}}, \beta)$, where $\theta^*_{\text{SGD}}$ is typically an SGD solution from a standard training workflow. In their method, the optimal $\beta$ is computed *post hoc* by optimizing a similar objective, $\arg\max_\beta \frac{1}{n} \sum_{(x,y) \in \mathcal{D}_{\text{valid}}} p(y|x, \theta^*_{\text{SGD}}, \beta)$, using a validation set, with $\theta$ fixed and only $\beta$ being optimized. While the difference might seem subtle, we find that jointly optimizing $\theta$ and $\beta$ is the key to obtaining a good $\beta$ for constructing PPDs. Detailed experimental results are provided in Appendix D.

### 3.2 SUPPORTING THEORY

Ideally, we would like to theoretically show that the temperature selection method in (7) produces high test LPD of (3). However, it turns out that the theoretical guarantees are more natural for a related object which is suggested by the reformulation of the tempered posterior in (6). Namely, we can consider an alternative PPD that is the expectation of the *tempered model*:

$$\mathbb{E}_{p_\beta(\theta|\mathcal{D})} p(y|x, \theta, \beta) = \int p(y|x, \theta, \beta) p_\beta(\theta|\mathcal{D}) \mathrm{d}\theta. \tag{8}$$

This is also the object of study in Adlam et al. (2020) and is to be contrasted with the PPD in (3). We emphasize that our temperature selection method can be used with either (3) or (8), and we will compare the resulting performance of our method for these two PPDs in the experiments in Section 4.

To study the theoretical properties of the temperature selection method for (8), we consider the objective at the population level, leading to

$$\theta^*, \beta^* := \arg\max_{\theta, \beta} \mathbb{E}_{q(x,y)} \log p(y|x, \theta, \beta).$$

We justify $\beta^*$ in the case of Gaussian linear regression, but our temperature methodology can be applied in far more general settings as we demonstrate in Section 4. We first compute the test LPD constructed with (8) and show that $\beta^*$ approximately maximizes a lower bound of the test LPD.

**Lemma 3.1.** *Consider a linear regression model $p(y|x, \theta) = \mathcal{N}(y|x^\top \theta, \sigma^2)$ with a $d$-dimensional input $x$ and known variance $\sigma^2$, and a prior $p(\theta) = \mathcal{N}(\theta|0, \sigma_p^2)$ with finite variance $\sigma_p^2$. Let $\boldsymbol{X} := (x_1, \ldots, x_n)^\top \in \mathbb{R}^{n \times d}$ and $\boldsymbol{\Sigma} := (\boldsymbol{X}^\top \boldsymbol{X} + \frac{\sigma^2}{\sigma_p^2} \boldsymbol{I})^{-1}$. The test LPD of the PPD in (8) at a fixed $\beta$ is bounded from below:*

$$\mathrm{LPD}(\mathbb{E}_{p_\beta(\theta|\mathcal{D})}[p(y|x, \theta, \beta)]) > \mathbb{E}_{q(x,y)} \log p(y|x, \hat{\theta}_{MAP}, \beta) - \frac{1}{2} \mathbb{E}_{q(x,y)} \log(1 + x^\top \boldsymbol{\Sigma} x),$$

*where $\hat{\theta}_{MAP} := \boldsymbol{\Sigma} \boldsymbol{X}^\top \boldsymbol{y}$ is the maximum-a-posteriori solution of the posterior $p_\beta(\theta|\mathcal{D})$ at $\beta = 1$ and $\boldsymbol{y} := (y_1, \ldots y_n)^\top \in \mathbb{R}^n$.*

The proof can be found in Appendix E. As our goal is to select $\beta$ that maximizes test LPD, a reasonable strategy is to optimize the lower bound presented in Lemma 3.1 with respect to $\beta$. As the second term in the lower bound is independent of $\beta$, this is therefore equivalent to maximizing the first term, $\mathbb{E}_{q(x,y)} \log p(y|x, \hat{\theta}_{\text{MAP}}, \beta)$. Note that this objective function requires $\hat{\theta}_{\text{MAP}}$, which is often unavailable in closed-form. A straightforward solution is to replace it with an estimate from an iterative optimizer, before optimizing again with respect to $\beta$. This comes at the cost of two optimization runs (and the effort to tune their hyperparameters). Instead, we propose maximizing the empirical version of $\mathbb{E}_{q(x,y)} \log p(y|x, \theta, \beta)$ with respect to $\theta$ and $\beta$ simultaneously, leading us back to (7). We study the efficacy of this method empirically in the next section.

## 4 EXPERIMENTS

We now illustrate the behavior of the PPDs, (3) and (8), across different values of $\beta$ on a suite of benchmark datasets for both regression and classification tasks. We refer to (3) as SM-PD and

(8) as TM-PD, where SM and TM stand for standard model and tempered model, respectively. Notably, our data-driven procedure for selecting $\beta$ is agnostic to its downstream use in either SM-PD or TM-PD. We demonstrate that the proposed method performs well in both according to the test LPD metric. While our method aims to maximize test LPD and is supported by the theory discussed in Section 3.2, we also evaluate the point prediction of the PPD. Specifically, we use $\hat{y} = \mathbb{E}_{p_\beta(y|x,\mathcal{D})}[y]$ to compute mean squared error (MSE) $\frac{1}{|\mathcal{D}_{\text{test}}|} \sum_{(x,y)\in\mathcal{D}_{\text{test}}} (y - \hat{y})^2$ for regression tasks, and $\hat{y} = \arg\max_y p_\beta(y|x,\mathcal{D})$ to compute accuracy $\frac{1}{|\mathcal{D}_{\text{test}}|} \sum_{(x,y)\in\mathcal{D}_{\text{test}}} \mathbb{1}[y = \hat{y}]$ for classification tasks. By definition, these point predictions $\hat{y}$ are identical across SM-PD and TM-PD, and their results are consolidated in our reporting. We compare our method against a grid search over nine $\beta \in \{0.1, 0.3, 1, 3, \ldots, 1000\}$, corresponding to increments of roughly 0.5 on the $\log_{10}$ scale. The optimal $\beta$ from the grid search is selected based on test LPD, MSE, or accuracy on a validation set[2], and the optimal $\beta$ may differ depending on the metric used.

To illustrate our method, we follow the experimental setup of Wenzel et al. (2020). Regression tasks are conducted using a one-layer ReLU network on UCI datasets (Concrete, Energy, Naval), while classification is performed using a CNN on MNIST and a ResNet20 on CIFAR10, both with and without data augmentation (DA). For the prior on neural network weights, we restrict $p(\theta)$ to a zero-mean isotropic Gaussian, as this is a common choice for achieving state-of-the-art performance (Izmailov et al., 2021). Further details on the model and prior can be found in Appendix G.

We employ the stochastic-gradient Markov chain Monte Carlo (SGMCMC) algorithm from (Wenzel et al., 2020) with a cyclical step-size scheduler (Zhang et al., 2020) to sample from the posterior (details in Appendix H). Hyperparameters are carefully tuned based on the temperature diagnostics in Wenzel et al. (2020) to ensure sampler convergence to the posterior with the specified $\beta$; see Appendix I and Appendix J for diagnostics and hyperparameters. This approach contrasts with the common practice of tuning hyperparameters for predictive performance. For CIFAR10, we collect 30 samples per run, and 100 for all other datasets. Each SGMCMC run is repeated five times with different initializations to generate five sets of posterior samples.

## 4.1 RESULTS

**Optimal temperature of SM-PD and TM-PD across models and datasets.** The test LPD across different temperatures for SM-PD and TM-PD are shown in Figure 1; see Appendix K for additional figures and Table 8 and Table 9 for tabular versions of the results. For Concrete, Energy, Naval and MNIST, we observe that TM-PD generally outperforms SM-PD. This is perhaps not surprising, as we expect the former to better account for aleatoric uncertainty. For Concrete and MNIST, the peaks of SM-PD and TM-PD also coincide. Surprisingly, TM-PD does not outperform SM-PD in the CIFAR10 examples.

In terms of the efficacy of our temperature selection method, we find that it can generally recover $\beta$ with a good test LPD. An interesting observation here is that our method tends to select the optimal $\beta$ for TM-PD in the regression examples, and optimal $\beta$ for SM-PD in the classification examples. Moreover, as shown in Figure 1, the optimal $\beta$ for TM-PD and SM-PD for a given model are also not always the same. Therefore, we expect that the $\hat{\beta}^*$, which is computed by optimizing (7) irrespective of the construction of the PPD, may only work for either TM-PD or SM-PD, but not both at the same time.

We also report MSE (for regression) and accuracy (for classification) to assess the point predictions of the PPDs. As the point predictions are identical across SM-PD and TM-PD by definition, we consolidate the results in Table 1. In general, we observe that SM-PD and TM-PD outperform both SGD and the PPD at $\beta = 1$ across both metrics. Furthermore, our method can select temperatures that achieves performance comparable to that of the grid search.

**Optimal temperature across data augmentation strength.** It has been frequently observed that data augmentation is one of the key ingredients for observing CPE, and the optimal temperature may depend on the strength of data augmentation (Bachmann et al., 2022). In view of this, we conducted an experiment to determine how the optimal temperature differs across different levels of augmentation. Here we only focus on SM-PD and present the results in Figure 2. We observe

---

[2]In contrast, our method tracks the log-likelihood of $p(y|x, \theta, \beta)$ on a validation set

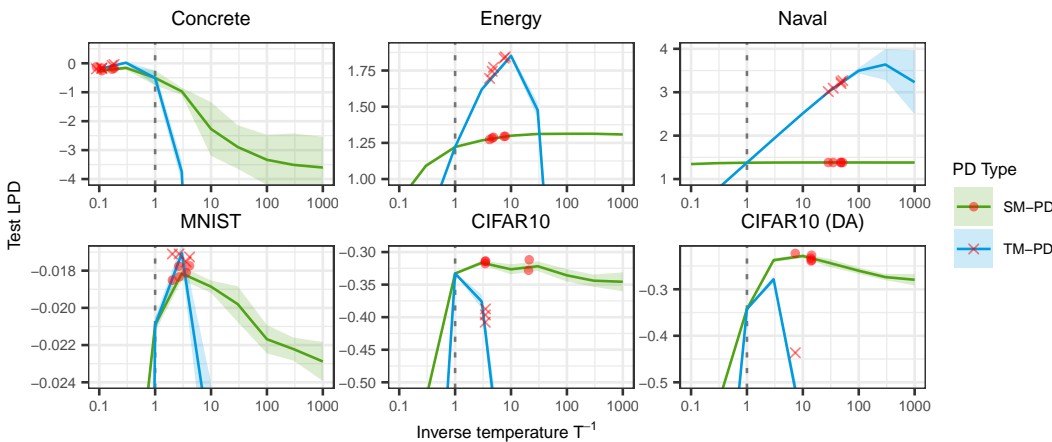

Figure 1: Test LPD plotted against inverse temperature $\beta$. We compare two types of PPD: SM-PD (green) as defined in (3) and TM-PD (blue) as defined in (8). Zoomed-in versions of these curves are also provided in Figure 4 and Figure 5, respectively. In each example, we have five evaluations of $\hat{\beta}^*$ from our method. Each of these $\hat{\beta}^*$ has a corresponding test LPD computed with SM-PD (red circle) and TM-PD (red cross). Some of the red crosses in the CIFAR-10 examples are out of range. Solid lines and shaded areas represent the mean $\pm$ standard error across five repetitions. The vertical dotted lines indicate the PPD at $\beta = 1$. Higher test LPD is better.

Table 1: MSE (for regression) and accuracy (for classification) of the point predictions of the PPDs at $\beta = 1$, $\beta = \hat{\beta}^*$ (our method), and the optimal $\beta$ obtained from grid search. We also include the predictions from SGD as a baseline. The results for SM-PD and TM-PD are consolidated, as both produce identical point predictions by definition. The presented values are means $\pm$ standard error across five repetitions, with the best value among the four methods boldfaced. The PPDs at $\beta \neq 1$ generally outperform SGD and the PPD at $\beta = 1$, and our method can achieve performance comparable to that of grid search.

| METHOD | MSE $\downarrow$ ($\times 10^{-3}$) | | | ACCURACY $\uparrow$ | | |
|---|---|---|---|---|---|---|
| | CONCRETE | ENERGY | NAVAL | MNIST | CIFAR10 | CIFAR10 (DA) |
| SGD | $106 \pm 20$ | $1.7 \pm 0.2$ | $0.149 \pm 0.063$ | $99.05 \pm 0.11$ | $84.9 \pm 3.4$ | $91.9 \pm 0.3$ |
| $\beta = 1$ | $\mathbf{75 \pm 7}$ | $2.1 \pm 0.1$ | $0.045 \pm 0.005$ | $99.28 \pm 0.04$ | $89.1 \pm 0.3$ | $88.4 \pm 0.2$ |
| $\beta = \hat{\beta}^*$ | $82 \pm 4$ | $1.6 \pm 0.1$ | $0.032 \pm 0.009$ | $\mathbf{99.38 \pm 0.03}$ | $\mathbf{89.9 \pm 0.2}$ | $\mathbf{92.8 \pm 0.2}$ |
| GRID | $76 \pm 8$ | $\mathbf{1.4 \pm 0.1}$ | $\mathbf{0.027 \pm 0.008}$ | $99.32 \pm 0.05$ | $\mathbf{89.9 \pm 0.2}$ | $\mathbf{92.8 \pm 0.4}$ |

a subtle and gradual increase in the optimal temperature with the strength of data augmentation. Moreover, our method can also recover temperatures that produce good test LPD and accuracy under different augmentation levels. Note that, in addition to the well-known CPE observed in CIFAR-10 with data augmentation, it also manifests in a milder form in our CIFAR-10 experiments without data augmentation. This contrasts with previous studies reporting the absence of CPE in the same setup (Izmailov et al., 2021).

**Computation time**   Our approach requires a single SGD run to compute the optimal $\beta$ and a SGMCMC run to generate the PPD, whereas the grid search requires 9 SGMCMC runs — one for each temperature in the grid — to identify the best PPD. The wall-clock times for the main experiment are reported in Table 2. Overall, our method is 4 times faster than the grid search for the smaller regression models and 8 times faster for the larger classification models.

Table 2: Wall-clock time comparison across methods. Our method is about 4 times faster than the grid search (of 9 temperatures) for the smaller regression models, and 8 times faster for the larger classification models.

| METHOD | WALL-CLOCK TIME (MINUTES) | | | | | |
|---|---|---|---|---|---|---|
| | CONCRETE | ENERGY | NAVAL | MNIST | CIFAR10 | CIFAR10 (DA) |
| SGD | 0.17 | 0.16 | 0.78 | 1.93 | 27.90 | 34.84 |
| $\beta = 1$ | 0.20 | 0.19 | 1.40 | 18.83 | 260.52 | 327.12 |
| $\beta = \hat{\beta}^*$ (OURS) | 0.37 | 0.36 | 2.19 | 20.77 | 288.42 | 361.96 |
| GRID | 1.76 | 1.74 | 12.64 | 169.51 | 2344.15 | 2944.01 |

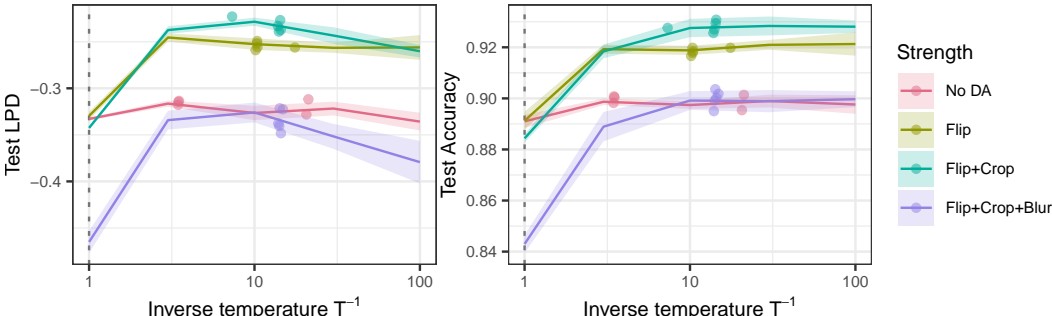

Figure 2: Test LPD and accuracy of CIFAR-10 plotted against inverse temperature $\beta$ under various levels of data augmentation (color). The lines and shaded areas represent the mean $\pm$ standard error across five repetitions. There are five dots for each colored curve, and each of these dots corresponds to a repetition of $\hat{\beta}^*$ from our method. The vertical dotted lines indicate the PPD at $\beta = 1$. There is a subtle shift of peaks from left to right as the augmentation strength increases. Higher test LPD and accuracy indicate better performance.

## 5  CPE THROUGH THE LENS OF BAYESIAN DEEP LEARNING

Since the publication of Wenzel et al. (2020), there have been numerous attempts to explain CPE, with particular focus on predictive metrics such as test LPD, accuracy (for classification), and MSE (for regression). In addition to the arguments presented in Section 3, we provide a summary of the popular insights into CPE from the BDL literature here.

**Poor posterior approximation.**    One explanation for the presence or absence of CPE is inadequate posterior approximation, e.g., an inappropriate choice of step size, or the omission of Metropolis-Hastings steps in the SGMCMC sampler. However, as shown in Appendices A and B, CPE is not solely an artifact of poor posterior approximation and can be observed theoretically.

**Likelihood corruption due to data augmentation.**    It has been widely observed that the use of data augmentation often amplifies CPE (Izmailov et al., 2021), while being less pronounced when data augmentation is turned off. It is argued that the augmented data violate the usual independent and identically distributed (i.i.d.) assumption imposed on the data. However, CPE has been found to persist even after accounting for this assumption violation (Nabarro et al., 2022). Therefore, CPE is unlikely to be a mere artifact of data augmentation. In a separate analysis, Kapoor et al. (2022) argues that the SGMCMC sampler will converge to a tempered posterior in the presence of data augmentation. They concluded that the likelihood is implicitly raised to a power equal to the number of augmentations, and raising the likelihood to a power reciprocal of this number should approximately recover the standard posterior. However, they do not find this adjustment alone sufficient to remove CPE completely.

**Model misspecification.** It has been argued that data augmentation and curation may lead to model misspecification, in particular overestimating the aleatoric uncertainty in the data (Aitchison, 2021; Kapoor et al., 2022). Therefore, tempering is proposed to be an effective tool for correction. This finding aligns with Bachmann et al. (2022), which showed the optimal temperature dependent on both the aleatoric uncertainty in the data and the 'invariance' of the model to augmented data.

**Prior misspecification.** While the exact interpretation of a 'well-specified prior' is debatable, Fortuin et al. (2022) conducted a large-scale experiment to study the effect of tempering under Gaussian (isotropic or with correlated covariance) and heavy-tailed priors (Student's-t or Laplace) on four metrics: test LPD, accuracy, expected calibration error (Naeini et al., 2015), and out-of-distribution detection accuracy. They observed that $\beta = 1$ is indeed optimal for test LPD and accuracy when using a heavy-tailed prior. However, the standard posteriors derived from heavy-tailed priors also tend to underperform compared to tempered posteriors derived from Gaussian priors. Notably, according to their experiments, warmer temperatures tend to perform better for expected calibration error, and there is no general trend in optimal $\beta$ for out-of-distribution detection accuracy. Therefore, their work suggests that the optimal $\beta$ is not only dependent on the truth-model-prior triplet but also on the evaluation metric.

**Attempts to 'fix' the CPE.** Within much of the CPE literature, tempering is often seen as a 'hack' that strays from the Bayesian principle. This has prompted the development of various 'fixes' — models and priors that induce standard posteriors with similar predictive performance to tempered posteriors (see, e.g., Fortuin et al., 2022; Kapoor et al., 2022; Marek et al., 2024). However, we argue that this is unnecessary, as tempered posteriors of arbitrary $\beta > 0$ are special cases of the generalized Bayes posterior (Bissiri et al., 2016); see Appendix L for a primer. Therefore, posterior tempering does not deviate from the Bayesian principle.

## 6 CPE THROUGH THE LENS OF GENERALIZED BAYES

In GB, it is a standard practice to tune $\beta$ to improve the utility of the tempered posterior (Zhang, 2006a; Grünwald, 2012; Bissiri et al., 2016). Specifically, the likelihood is frequently (but not always) tempered to a warmer temperature, which contrasts with the cold temperature used in BDL. In this section, we review some recent developments in GB to clarify the role of tempering in GB and this apparent contradiction.

**Ensuring posterior concentration on the KL minimizer.** Posterior concentration on the KL minimizer $\theta_\dagger = \arg\min_\theta D_{\mathrm{KL}}(q(y|x)|p(y|x,\theta))^3$ as $n \to \infty$ is often a desired property for many statistical applications. Proofs establishing this property (Barron & Cover, 1991; Zhang, 2006a; Grünwald, 2007) typically assume that the following inequality holds for all $\theta$ in the parameter space:

$$\mathbb{E}_{q(x,y)}\left[\left(\frac{p(y|x,\theta)}{p(y|x,\theta_\dagger)}\right)^\beta\right] \leq 1, \quad \text{for all } \theta \in \Theta.$$

It is easy to verify that the inequality holds at $\beta = 1$ for well-specified models, where $q(x,y) = p(y|x,\theta_\dagger)q(x)$. The same cannot be said for misspecified models in general. However, Grünwald (2012) shows that this inequality can still hold for many misspecified models for some warm temperatures $\beta \leq \beta_{\mathrm{critical}} < 1$, and proposed the SafeBayes algorithm to determine $\beta_{\mathrm{critical}}$. Therefore, SafeBayes is presented as a tool to achieve posterior concentration on $\theta_\dagger$.

However, posterior concentration is not always desirable in the context of maximizing test LPD under model misspecification. Note that taking $\beta \leq \beta_{\mathrm{critical}}$ results in the corresponding PPD concentrating on the 'plug-in' predictive density $p(y|x,\theta_\dagger)$ as $n \to \infty$. This may be undesirable if one prioritizes the test LPD of the PPD, since the plug-in $p(y|x,\theta_\dagger)$ may not exhibit good test LPD. For example, when the misspecified model is non-convex, the PPD can lie outside the model family and be closer to the truth than the plug-in $p(y|x,\theta_\dagger)$. Therefore, avoiding posterior concentration in this situation can actually result in a PPD with a higher test LPD.

---

[3]In GB, the minimizer is assumed to be unique, though this is often not true for neural network models.

**Calibrating credible and prediction intervals.** Constructing well-calibrated credible and prediction intervals with the nominal frequentist coverage probability is known to be challenging, as obtaining the posterior variance of a misspecified model is difficult (Kleijn & van der Vaart, 2012, Example 2.1). These issues can be mitigated by appropriately tuning $\beta$, which affects the spread of the posterior. To this end, Syring & Martin (2019) and Wu & Martin (2021) developed algorithms to select $\beta$ for calibrating credible and prediction intervals, respectively.

**Calibrating prior-to-posterior information gain.** In decision-theoretic GB (Bissiri et al., 2016, see Appendix L for a primer), tempered posteriors (with likelihood tempering only) are seen as an update rule that combines data information with prior belief. From this perspective, $\beta$ can be interpreted as a 'learning rate' at which information is 'transferred' to the posterior. Hence, it is reasonable to calibrate the information gain at each update. Temperature selection algorithms that follow this approach include Holmes & Walker (2017) and Lyddon et al. (2019).

**Improving test LPD.** Concurrent to our work, McLatchie et al. (2024) analyzed the role of $\beta$[4] in improving test LPD theoretically. They concluded that, in the moderately large $n$ regime and assuming posterior concentration at $\theta_\dagger$, the test LPD shows little improvement once $\beta$ becomes sufficiently large. Although their motivation aligns with ours, their theoretical results do not extend to singular models (e.g., neural networks) or over-parameterized models (e.g., ResNet20 on CIFAR10).

## 6.1 LIMITATIONS OF THE GB FRAMEWORK FOR ANALYZING BDL MODELS

The existing theory in GB, though elegant, is limited to regular and under-parameterized models and cannot fully account for CPE observed in BDL. Moreover, the metrics emphasized by GB often differ from those prioritized by BDL practitioners, such as test LPD. As a result, GB works do not provide a prescriptive methodology for selecting an appropriate temperature in BDL. Additionally, many current temperature selection algorithms (Grünwald, 2012; Syring & Martin, 2019; Wu & Martin, 2021) require repeated posterior computations, making them impractical for modern deep learning models. Finally, the GB literature primarily considers posteriors with likelihood tempering only, i.e., leaving the prior without tempering. However, since many priors $p(\theta)$ encountered in BDL are proper and bounded, prior tempering can be regarded as rescaling the prior (Kapoor et al., 2022, Section C.2). Therefore, many of the arguments in GB still apply.

## 7 CONCLUSION

In this work, we proposed a data-driven approach for selecting a good $\beta$ for use in either of the PPDs (3) and (8). Our approach circumvents the costly grid search method, i.e., sampling posterior at each $\beta$ across the grid, by optimizing a likelihood function (7) to obtain a good $\beta$. The $\hat{\beta}^*$ obtained via our method was shown to achieve comparable test LPD to that obtained from a grid search, all without performing any extra posterior sampling.

Additionally, we presented a detailed discussion to address the seeming contradiction in the optimal $\beta$ recommendations from the BDL and GB communities. We concluded that the optimal $\beta$ can differ depending on the specific downstream task.

**Limitation.** Our method is subject to several limitations that require future attention. Firstly, a poorly-tuned optimization procedure may result in a poor estimation of $\hat{\beta}^*$. Secondly, our method has only been empirically tested with the ubiquitous Gaussian priors. Lastly, while our method is empirically effective for modern neural network models and theoretically justified for Gaussian linear regression, a formal guarantee for general neural network models is still lacking.

**Future directions.** We hope that this work will inspire further research on data-driven approaches to select $\beta$, aimed not only at minimizing test LPD but also at optimizing other metrics, such as expected calibration error.

---

[4]In McLatchie et al. (2024), the *learning rate* and, confusingly, *temperature* refer to $\beta$ in this work.

## 8 REPRODUCIBILITY STATEMENT

The experiments details are summarized in the introduction of Section 4. Proof of Lemma 3.1 is given in Appendix E. SGMCMC is described in Appendix H. Diagnostic check of the SGMCMC is given in Appendix I. Implementation details of our proposed method is given in Appendix C. The set of hyperparameters to reproduce the experiments is given in Appendix J. Computational environment and packages are given in Appendix J.1. Evaluation metrics are defined in Appendix F. Model, prior and datasets are described in Appendix G. Source code is published on `https://anonymous.4open.science/r/tempered-posteriors-106E`.

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

# A    OPTIMAL TEMPERATURE OF A GAUSSIAN TOY MODEL

In this section, we compute the optimal temperature that minimizes the 2-Wasserstein distance and the KL divergence between the truth and PPD of a toy Gaussian model. We show that: 1) the optimal temperature depends on both the evaluation metric and the truth-model-prior triplet, and 2) the optimal temperature that maximizes test LPD is not necessarily 1, even in the case of a well-specified model.

In our setup, we assume a set of i.i.d. samples $\mathcal{D} = \{x_i\}_{i=1}^n$ drawn from a univariate Gaussian truth $q(x) = \mathcal{N}(x|0, \tau^2)$ with a known variance $\tau^2$. Our model is given by $p(x|\mu) = \mathcal{N}(x|\mu, \sigma^2)$ with a fixed variance $\sigma^2$. We also assume a Gaussian prior on $\mu$, i.e., $p(\mu) = \mathcal{N}(\mu|0, \sigma_p^2)$. While we focus on the unsupervised setting for the ease of exposition, our argument also applies to the supervised settings.

The tempered posterior at $\beta = \frac{1}{T}$ is given by:

$$
p_\beta(\mu|\mathcal{D}) \propto p(\mu)^{1/T} \prod_{i=1}^n \mathcal{N}(x_i|\mu, \sigma^2 T)
$$

$$
= \mathcal{N}\left(\mu \,\middle|\, n\bar{x}\frac{\sigma_{post}^2}{\sigma^2 T}, \sigma_{post}^2\right)
$$

$$
= \mathcal{N}\left(\mu \,\middle|\, \bar{x}\left(\frac{\sigma^2}{n\sigma_p^2} + 1\right)^{-1}, \sigma_{post}^2\right),
$$

where the posterior variance is given by $\sigma_{post}^2 = T\left(\frac{1}{\sigma_p^2} + \frac{n}{\sigma^2}\right)^{-1}$.

The PPD as defined in (3) is then:

$$
p_\beta(x|\mathcal{D}) = \int \mathcal{N}(x|\mu, \sigma^2) p_\beta(\mu|\mathcal{D}) \mathrm{d}\mu = \mathcal{N}\left(x \,\middle|\, n\bar{x}\frac{\sigma_{post}^2}{\sigma^2 T}, \sigma_{post}^2 + \sigma^2\right)
$$

## A.1    OPTIMAL TEMPERATURE FOR MINIMIZING 2-WASSERSTEIN DISTANCE

As both $q(x)$ and $p_\beta(x|\mathcal{D})$ are Gaussian, the 2-Wasserstein distance $W_2$ between these two density has a closed-form expression:

$$
W_2 = \left(\left(\left(\frac{\sigma^2}{n\sigma_p^2} + 1\right)^{-1}\bar{x}\right)^2 + T\left(\frac{1}{\sigma_p^2} + \frac{n}{\sigma^2}\right)^{-1} + \sigma^2 + \tau^2 - 2\tau\left[T\left(\frac{1}{\sigma_p^2} + \frac{n}{\sigma^2}\right)^{-1} + \sigma^2\right]\right)^{1/2}
$$

Taking the gradient of $W_2$ with respect to $T$ and set it to 0, we can derive the optimal temperature $T^*$:

$$
\frac{\partial W_2}{\partial T} = \left(\frac{1}{\sigma_p^2} + \frac{n}{\sigma^2}\right)^{-1} - \tau\left[T\left(\frac{1}{\sigma_p^2} + \frac{n}{\sigma^2}\right)^{-1} + \sigma^2\right]^{-1/2}\left(\frac{1}{\sigma_p^2} + \frac{n}{\sigma^2}\right)^{-1} = 0
$$

$$
\implies T^* = (\tau^2 - \sigma^2)\left(\frac{1}{\sigma_p^2} + \frac{n}{\sigma^2}\right), \quad T^* \in (0, \infty)
$$

Therefore, we can conclude that the optimal $T^* \to 0$ in the well-specified case ($\tau^2 = \sigma^2$).

## A.2 Optimal Temperature for Minimizing KL Divergence

We can also obtain the closed-form expression KL divergence, $D_{\mathrm{KL}}(q(x)\|p_\beta(x|\mathcal{D}))$:

$$
\begin{aligned}
D_{\mathrm{KL}}(q(x)\|p_\beta(x|\mathcal{D})) &= \mathbb{E}_{q(x)}\left[\log\frac{q(x)}{p_\beta(x|\mathcal{D})}\right] \\
&= \left[\left(\frac{\sigma^2}{n\sigma_p^2}+1\right)^{-1}\bar{x}\right]^2 \frac{1}{2(\sigma_{post}^2+\sigma^2)} \\
&\quad + \frac{1}{2}\left(\frac{\tau^2}{\sigma_{post}^2+\sigma^2}-1-\ln\frac{\tau^2}{\sigma_{post}^2+\sigma^2}\right).
\end{aligned}
$$

Note that this object directly corresponds to test log predictive density (LPD), which is the primary object of interest in Bayesian deep learning and our work. Taking the derivative of the divergence with respect to $T$ and set it to 0, we can solve for the posterior variance that minimizes the divergence:

$$
\frac{\partial D_{\mathrm{KL}}}{\partial T} = \left[-\left[\left(\frac{\sigma^2}{n\sigma_p^2}+1\right)^{-1}\bar{x}\right]^2 \frac{1}{(\sigma_{post}^2+\sigma^2)^2} + \left(\frac{-\tau^2}{(\sigma_{post}^2+\sigma^2)^2}+\frac{1}{\sigma_{post}^2+\sigma^2}\right)\right]\frac{1}{2}\frac{\partial\sigma_{post}^2}{\partial T} = 0
$$

$$
\implies \sigma_{post}^2 = \left[\left(\frac{\sigma^2}{n\sigma_p^2}+1\right)^{-1}\bar{x}\right]^2 + \tau^2 - \sigma^2
$$

From this posterior variance, we can deduce the optimal $T^*$:

$$
\begin{aligned}
T^* &= \left[\left[\left(\frac{\sigma^2}{n\sigma_p^2}+1\right)^{-1}\bar{x}\right]^2 + \tau^2 - \sigma^2\right]\left(\frac{1}{\sigma_p^2}+\frac{n}{\sigma^2}\right) \\
&= \left(\frac{n}{\sigma^2}\bar{x}\right)^2\left(\frac{1}{\sigma_p^2}+\frac{n}{\sigma^2}\right)^{-1} + (\tau^2-\sigma^2)\left(\frac{1}{\sigma_p^2}+\frac{n}{\sigma^2}\right), \quad T^* \in (0,\infty).
\end{aligned}
$$

We can see that $T^*$ is non-trivial, and $T^* \neq 1$ in the well-specified case ($\tau^2 = \sigma^2$). This contradicts with the popular belief that the test LPD is maximized by the PPD induced from a well-specified standard posterior (Adlam et al., 2020).

## B Optimality Conditions for Test LPD at $\beta = 1$

In this section, we follow the work of Zhang et al. (2024) (which assumes likelihood tempering rather than full posterior tempering) and argue that the PPD (3) is rarely optimal in terms of test LPD at $\beta = 1$. Our approach is to demonstrate that the gradient $\nabla_\beta \mathrm{LPD}(p_\beta(y|x,\mathcal{D}))$ is rarely zero at $\beta = 1$.

**Lemma B.1.** *Let $p(\mathcal{D},\theta) = p(\theta)\prod_{(x,y)\in\mathcal{D}}p(y|x,\theta)$, where we suppress the conditional dependency on $x$ in $p(\mathcal{D},\theta)$. Then, the gradient of $\mathrm{LPD}(p_\beta(y|x,\mathcal{D}))$ with respect to $\beta$ is given by*

$$
\nabla_\beta \mathrm{LPD}(p_\beta(y|x,\mathcal{D})) = \mathbb{E}_{q(x,y)}\left[\mathbb{E}_{p_\beta(\theta|\mathcal{D}\cup(x,y))}[\log p(\mathcal{D},\theta)]\right] - \mathbb{E}_{p_\beta(\theta|\mathcal{D})}[\log p(\mathcal{D},\theta)],
$$

*where $p_\beta(\theta|\mathcal{D}\cup(x,y)) \propto p(y|x,\theta)p_\beta(\theta|\mathcal{D})$ represents an update to the posterior $p_\beta(\theta|\mathcal{D})$ after observing an extra data point $(x,y)$ from the truth, and the expectation $\mathbb{E}_{q(x,y)}[\cdot]$ integrates over $(x,y)$ in $p(\theta|\mathcal{D}\cup(x,y))$.*

*Proof.* Let $\mathbb{E}_{p_\beta}[\cdot] := \mathbb{E}_{p_\beta(\theta|\mathcal{D})}[\cdot]$ represent the expectation with respect to the tempered posterior $p_\beta(\theta|\mathcal{D})$, and let the normalizing constant of the tempered posterior be $Z(\mathcal{D},\beta) := \int p(\mathcal{D},\theta)^\beta \mathrm{d}\theta$.

We begin by deriving an identity that will be useful for the gradient computation. For any arbitrary $f : \mathbb{R}^d \to \mathbb{R}$, the following identity holds:

$$\nabla_\beta \mathbb{E}_{p_\beta} f(\theta) = \int \nabla_\beta [f(\theta) p_\beta(\theta|\mathcal{D})] \mathrm{d}\theta$$

$$= \int f(\theta) \nabla_\beta \log p_\beta(\theta|\mathcal{D}) p_\beta(\theta|\mathcal{D}) \mathrm{d}\theta$$

$$= \mathbb{E}_{p_\beta}[f(\theta) \nabla_\beta \log p_\beta(\theta|\mathcal{D})]$$

$$= \mathbb{E}_{p_\beta}[f(\theta) \nabla_\beta \log p(\mathcal{D}, \theta)^\beta] - \mathbb{E}_{p_\beta}[f(\theta)] \nabla_\beta \log Z(\mathcal{D}, \beta)$$

$$= \mathbb{E}_{p_\beta}[f(\theta) \log p(\mathcal{D}, \theta)] - \mathbb{E}_{p_\beta}[f(\theta)] \mathbb{E}_{p_\beta}[\log p(\mathcal{D}, \theta)],$$

where $\nabla_\beta \log Z(\mathcal{D}, \beta)$ in the second last line becomes

$$\nabla_\beta \log Z(\mathcal{D}, \beta) = \frac{\int \nabla_\beta p(\mathcal{D}, \theta)^\beta \mathrm{d}\theta}{Z(\mathcal{D}, \beta)} = \int \log p(\mathcal{D}, \theta) \frac{p(\mathcal{D}, \theta)^\beta}{Z(\mathcal{D}, \beta)} \mathrm{d}\theta = \mathbb{E}_{p_\beta}[\log p(\mathcal{D}, \theta)].$$

The gradient can then be computed as follows:

$$\nabla_\beta \mathrm{LPD}(p_\beta(y|x, \mathcal{D})) = \nabla_\beta \mathbb{E}_{q(x,y)} \log \mathbb{E}_{p_\beta} p(y|x, \theta)$$

$$= \mathbb{E}_{q(x,y)} \frac{\nabla_\beta \mathbb{E}_{p_\beta} p(y|x, \theta)}{\mathbb{E}_{p_\beta} p(y|x, \theta)}$$

$$= \mathbb{E}_{q(x,y)} \left[ \frac{\mathbb{E}_{p_\beta}[p(y|x, \theta) \log p(\mathcal{D}, \theta)]}{\mathbb{E}_{p_\beta} p(y|x, \theta)} \right] - \mathbb{E}_{p_\beta}[\log p(\mathcal{D}, \theta)]$$

$$= \mathbb{E}_{q(x,y)}[\mathbb{E}_{p_\beta(\theta|\mathcal{D} \cup (x,y))} \log p(\mathcal{D}, \theta)] - \mathbb{E}_{p_\beta}[\log p(\mathcal{D}, \theta)],$$

where the term inside $\mathbb{E}_{q(x,y)}[\cdot]$ in the third equality simplifies to

$$\frac{\mathbb{E}_{p_\beta}[p(y|x, \theta) \log p(\mathcal{D}, \theta)]}{\mathbb{E}_{p_\beta} p(y|x, \theta)} = \int \log p(\mathcal{D}, \theta) \frac{p(y|x, \theta) p_\beta(\theta|\mathcal{D})}{\int p(y|x, \theta') p_\beta(\theta'|\mathcal{D}) \mathrm{d}\theta'} \mathrm{d}\theta$$

$$= \mathbb{E}_{p_\beta(\theta|\mathcal{D} \cup (x,y))} \log p(\mathcal{D}, \theta).$$

$$\square$$

We can view the terms in the RHS of Lemma B.1 as a training loss under a posterior $\rho$:

$$\mathbb{E}_\rho L(\mathcal{D}, \theta) := \mathbb{E}_\rho[-\log p(\mathcal{D}, \theta)] = \mathbb{E}_\rho\left[-\log p(\theta) - \sum_{(x,y) \in \mathcal{D}} \log p(y|x, \theta)\right].$$

This is similar to the training loss as defined in Zhang et al. (2024), but with an additional regularizer $p(\theta)$ included in the loss. Then, we can define *underfitting* as $\mathbb{E}_{q(x,y)}[\mathbb{E}_{p_\beta(\theta|\mathcal{D} \cup (x,y))} L(\mathcal{D}, \theta)] <$ $\mathbb{E}_{p_\beta(\theta|\mathcal{D})} L(\mathcal{D}, \theta)$, i.e., the posterior $p_\beta(\theta|\mathcal{D} \cup (x,y))$ will, on average, have a lower training loss than the original $p_\beta(\theta|\mathcal{D})$ after receiving an extra observation $(x, y)$ from the truth. Similarly, we can define the converse as *overfitting*.

Therefore, it is not difficult to imagine that the posterior is rarely well-fitted at $\beta = 1$, and we should not expect optimality in terms of the test LPD from the PPD at $\beta = 1$.

## C   DETAILS OF THE TEMPERATURE SELECTION ALGORITHM

Our proposed procedure for constructing a PPD at an optimal temperature is summarized in Algorithm 1 with the following details:

1. We first run SGD to maximize the log-likelihood of the tempered model on a training set. We track the validation log-likelihood at the end of every $\lfloor L/20 \rfloor$-th epoch, where $L$ is the total number of epochs, to save computation time. We then select the temperature with the largest validation log-likelihood as our optimal temperature $\hat{\beta}^*$.

2. Subsequently, we run SGMCMC to draw samples from the tempered posterior at $\hat{\beta}^*$. We only start collecting samples after a burn-in phase. During the burn-in phase, we use a linear ramp function to control $T = \frac{1}{\beta}$ in (13), i.e. we first run SGMCMC at $T = 0$, then ramp up $T$ from 0 to $\hat{T}^* = \frac{1}{\hat{\beta}^*}$ and continue the burn-in at $\hat{T}^*$. This SGMCMC scheme is almost identical to Wenzel et al. (2020, Appendix A.1), except that we run extra burn-in epochs at $\hat{T}^*$.

---

**Algorithm 1:** Procedure to construct a PPD at the optimal temperature.

---

**Input:** Training data $\mathcal{D}$, Validation data $\mathcal{D}_{\text{valid}}$
Initialize $\theta \leftarrow$ some initializer, $\beta \leftarrow 1$;
Set $(\hat{\theta}^*, \hat{\beta}^*) \leftarrow (\theta, \beta)$;
**repeat**
  $\Delta \leftarrow$ Compute and scale $\nabla_{\theta, \log \beta} \sum_{(x,y) \in \mathcal{D}} \log p(y|x, \theta, \beta)$;
  $(\theta, \log \beta) \leftarrow (\theta, \log \beta) + \Delta$;
  **if** $\sum_{(x,y) \in \mathcal{D}_{\text{valid}}} \log p(y|x, \theta, \beta) > \sum_{(x,y) \in \mathcal{D}_{\text{valid}}} \log p(y|x, \hat{\theta}^*, \hat{\beta}^*)$ **then**
    $(\hat{\theta}^*, \hat{\beta}^*) \leftarrow (\theta, \beta)$
  **end**
**until** *Convergence or resource exhausted*;
$\mathcal{S} \leftarrow$ Draw $\theta$ from the tempered posterior $p_\beta(\theta|\mathcal{D})$ with SGMCMC at $T = \frac{1}{\hat{\beta}^*}$;
**Output:** The PPD $p_\beta(y|x, \mathcal{D}) \approx |\mathcal{S}|^{-1} \sum_{\theta \in \mathcal{S}} p(y|x, \theta)$ from the set of samples $\mathcal{S}$

---

## D    VARIANTS OF THE TEMPERATURE SELECTION METHOD

In this section, we compare a variant of (7) by solving

$$\arg\max_{\theta, \beta} \frac{1}{n} \sum_{(x,y) \in \mathcal{D}} [\log p(y|x, \theta, \beta) + \log p(\theta)]. \qquad (9)$$

We refer to this approach as the *maximum-a-posteriori* (MAP) method, as $\theta$ is now constrained by a prior $p(\theta)$. This contrasts with the *maximum likelihood* (MLE) method introduced in (7). Additionally, we compare our method to the popular temperature scaling approach proposed in Guo et al. (2017). In their method, $\theta$ in the objective (7) is fixed to a solution obtained from the standard training workflow (e.g., $\theta^*_{\text{SGD}}$), and the temperature $\beta$ is optimized by solving

$$\arg\max_{\beta} \frac{1}{n} \sum_{(x,y) \in \mathcal{D}_{\text{valid}}} \log p(y|x, \theta^*_{\text{SGD}}, \beta)$$

on a validation set. In contrast, both MLE and MAP jointly optimizes $\theta$ and $\beta$ within a single SGD run.

We compare the methods on the CIFAR-10 experiments, reporting the test LPD and accuracy in Table 3. The PPDs are constructed from SM-PD (3). In general, MAP and MLE yield similar $\beta$ values, while the $\beta$ obtained from Guo et al. (2017) differs substantially. Moreover, Guo et al. (2017) generally underperforms relative to MAP and MLE when the CPE is most pronounced, such as when data augmentation is enabled. This outcome is perhaps unsurprising, as Guo et al. (2017) is designed for computing a well-calibrated tempered model $p(y|x, \theta, \beta)$, whereas our method targets the PPD $p_\beta(y|x, \mathcal{D})$.

The results also suggest that optimizing $\theta$ and $\beta$ together is critical for determining a good $\beta$. We conjecture that $p(y|x, \theta^*, \beta^*)$, with both $\theta^*$ and $\beta^*$ obtained through our procedure, provides a reasonable approximation of the PPD with the highest test LPD. In contrast, fixing $\theta$ to a predetermined value (e.g., $\theta^*_{\text{SGD}}$) limits the search space and leads to a poor approximation of the PPD.

Another possible strategy here is the 'marginal likelihood method' which selects $\beta$ that maximizes the marginal likelihood of the tempered posterior. However, marginal likelihood is generally computationally intractable. Existing methods that rely on Laplace approximations Immer et al. (2021)

Table 3: Comparison between methods to select $\beta$ on the CIFAR10 experiments. Reported values are mean $\pm$ standard error across five repetitions. All values are evaluated on a test set. The accuracy and LPD are all within one standard error of difference between MAP and MLE.

| Data | Method | Accuracy ↑ | LPD ↑ | $\hat{\beta}^*$ |
|---|---|---|---|---|
| CIFAR10 | Guo et al. (2017) | $89.86 \pm 0.2$ | $-0.317 \pm 0.003$ | $1.75 \pm 0.18$ |
| | MAP | $89.89 \pm 0.25$ | $-0.321 \pm 0.010$ | $10.54 \pm 9.60$ |
| | MLE | $89.92 \pm 0.25$ | $-0.317 \pm 0.006$ | $10.44 \pm 9.54$ |
| CIFAR10 (DA) | Guo et al. (2017) | $90.47 \pm 0.14$ | $-0.286 \pm 0.004$ | $1.61 \pm 0.05$ |
| | MAP | $92.81 \pm 0.27$ | $-0.232 \pm 0.008$ | $14.28 \pm 0.52$ |
| | MLE | $92.81 \pm 0.20$ | $-0.232 \pm 0.007$ | $12.79 \pm 3.06$ |

are generally unreliable due to the singularity of neural network models Wei et al. (2022), although progress has been made to address this limitation (Hodgkinson et al., 2023).

# E  PROOF OF LEMMA 3.1

**Lemma E.1.** *Consider a linear regression model $p(y|x,\theta) = \mathcal{N}(y|x^\top\theta, \sigma^2)$ with a d-dimensional input $x$ and known variance $\sigma^2$, and a prior $p(\theta) = \mathcal{N}(\theta|0, \sigma_p^2)$ with finite variance $\sigma_p^2$. Let $\boldsymbol{X} :=$ $(x_1, \ldots, x_n)^\top \in \mathbb{R}^{n \times d}$ and $\boldsymbol{\Sigma} := (\boldsymbol{X}^\top \boldsymbol{X} + \frac{\sigma^2}{\sigma_p^2}\boldsymbol{I})^{-1}$. The test LPD of the PPD in (8) at a fixed $\beta$ is bounded from below:*

$$\text{LPD}(\mathbb{E}_{p_\beta(\theta|\mathcal{D})}[p(y|x,\theta,\beta)]) > \mathbb{E}_{q(x,y)}\log p(y|x, \hat{\theta}_{MAP}, \beta) - \frac{1}{2}\mathbb{E}_{q(x,y)}\log\left(1 + x^\top\boldsymbol{\Sigma}x\right),$$

*where $\hat{\theta}_{MAP} := \boldsymbol{\Sigma}\boldsymbol{X}^\top\boldsymbol{y}$ is the* maximum-a-posteriori *solution of the posterior $p_\beta(\theta|\mathcal{D})$ at $\beta = 1$ and $\boldsymbol{y} := (y_1, \ldots y_n)^\top \in \mathbb{R}^n$.*

*Proof.* Let $\sigma_\beta^2 = \frac{\sigma^2}{\beta}$. The tempered posterior can be shown to follow a Gaussian distribution

$$p_\beta(\theta|\mathcal{D}) \propto \mathcal{N}(\theta|0, \sigma_p^2/\beta) \prod_{x,y \in \mathcal{D}} \mathcal{N}(y|x^\top\theta, \sigma_\beta^2) \propto \mathcal{N}(\theta|\hat{\theta}, \sigma_\beta^2\boldsymbol{\Sigma}).$$

Therefore, the PPD (8) of linear regression can be derived using the identities of conditional Gaussian densities (see Bishop, 2006, Section 2.3.3)

$$
\begin{aligned}
\log\mathbb{E}_{p_\beta(\theta|\mathcal{D})}[p(y|x,\theta,\beta)] &= \log\int p(y|x,\theta,\beta)p_\beta(\theta|\mathcal{D})\mathrm{d}\theta \\
&= \log\int \mathcal{N}(y|x^\top\theta, \sigma_\beta^2)\mathcal{N}(\theta|\hat{\theta}, \sigma_\beta^2\boldsymbol{\Sigma})\mathrm{d}\theta \\
&= \log\mathcal{N}(y|x^\top\hat{\theta}, \sigma_\beta^2(1 + x^\top\boldsymbol{\Sigma}x)) \\
&= -\frac{1}{2}\log\left(1 + x^\top\boldsymbol{\Sigma}x\right) - \frac{1}{2}\log\left(2\pi\sigma_\beta^2\right) - \frac{(y - x^\top\hat{\theta})^2}{2\sigma_\beta^2(1 + x^\top\boldsymbol{\Sigma}x)} \\
&> -\frac{1}{2}\log\left(1 + x^\top\boldsymbol{\Sigma}x\right) - \frac{1}{2}\log\left(2\pi\sigma_\beta^2\right) - \frac{(y - x^\top\hat{\theta})^2}{2\sigma_\beta^2} \\
&= -\frac{1}{2}\log\left(1 + x^\top\boldsymbol{\Sigma}x\right) + \log\mathcal{N}(y|x^\top\hat{\theta}, \sigma_\beta^2) \\
&= -\frac{1}{2}\log\left(1 + x^\top\boldsymbol{\Sigma}x\right) + \log p(y|x, \hat{\theta}, \beta)
\end{aligned}
$$

Note that $\boldsymbol{\Sigma}$ is positive definite, and thus $1 + x^\top\boldsymbol{\Sigma}x > 1$ and the inequality holds. Taking expectation with respect to $q(x, y)$ at both sides of the inequality concludes the claim. $\qquad\square$

# F  EVALUATION METRICS

In addition to the LPD evaluated on a test set $\mathcal{D}_{\text{test}}$,

$$\mathbb{E}_{q(x,y)}[\log p_\beta(y|x, \mathcal{D})] \approx \frac{1}{|\mathcal{D}_{\text{test}}|}\sum_{x,y \in \mathcal{D}_{\text{test}}}\log p_\beta(y|x, \mathcal{D}),$$

we also report MSE for the regression examples

$$\frac{1}{|\mathcal{D}_{\text{test}}|}\sum_{x,y \in \mathcal{D}_{\text{test}}}(y - \hat{y})^2, \quad \hat{y} = \mathbb{E}_{p_\beta(y|x,\mathcal{D})}[y]$$

and accuracy for the classification examples

$$\frac{1}{|\mathcal{D}_{\text{test}}|}\sum_{x,y \in \mathcal{D}_{\text{test}}}\mathbb{1}[y = \hat{y}], \quad \hat{y} = \arg\max_y p_\beta(y|x, \mathcal{D})$$

where $\mathbb{1}$ is an indicator function. Note that the MSE and accuracy are indifferent to the constructions of PPD (3) and (8), as they both produce identical point predictions $\hat{y}$ in our setup.

Table 4: Details of the datasets, size of neural network and prior variance. We use the same prior for both CIFAR10 experiments with or without data augmentation.

|  | CONCRETE | ENERGY | NAVAL | MNIST | CIFAR10 |
|---|---|---|---|---|---|
| $\dim(x)$ | 8 | 8 | 14 | $28 \times 28 \times 1$ | $32 \times 32 \times 3$ |
| $\dim(\theta)$ | 641 | 641 | 1025 | 824458 | 273258 |
| $|\mathcal{D}|$ | 824 | 614 | 9547 | 60000 | 50000 |
| $|\mathcal{D}_{\text{valid}}|$ | 103 | 77 | 1194 | 5000 | 5000 |
| $|\mathcal{D}_{\text{test}}|$ | 103 | 77 | 1193 | 5000 | 5000 |
| PRIOR VARIANCE $\sigma_p^2$ | 0.1 | 0.1 | 1 | 0.1 | 1 |

## G  MODEL AND PRIOR DETAILS

### G.1  NETWORK ARCHITECTURE AND DATASETS

The network architecture and datasets are described in this section. The dimension of $x$ and $\theta$, and the sizes of the training, validation and testing sets are presented in Table 4.

**One-layer ReLU network on UCI datasets.**  The scalar mean function $\mu(\cdot; \theta)$ is parameterized with a 64-neuron hidden layer using ReLU activations. The datasets are Concrete (Yeh, 2007), Naval (Coraddu & Figari, 2014) and Energy (Tsanas & Xifara, 2012). They are all provided under the CC BY 4.0 license from UC Irvine Machine Learning Repository.

**CNN on MNIST.**  We utilize the convolutional neural network (CNN) provided in the MNIST example within the Flax tutorial. In broad terms, it comprises two convolutional layers (with 32 and 64 filters, respectively), followed by two fully connected layers (with 256 and 10 outputs). The convolutional layers employ $3 \times 3$ convolutions with ReLU activations and $2 \times 2$ average pooling. The MNIST dataset (LeCun et al., 2010) is provided under the CC BY-SA 3.0 license. Code for the CNN model can be found in `https://github.com/google/flax/blob/main/examples/mnist/train.py` under the Apache License, Version 2.0.

**ResNet20 on CIFAR10.**  We use the ResNet20 architecture (He et al., 2016) as implemented and ported from Wenzel et al. (2020). We also use the following data augmentation scheme, as in Wenzel et al. (2020):

- Random left and right flipping, then;

- Border-padding 4 zero values in both horizontal and vertical direction, followed by random cropping of the image to its original size.

The CIFAR10 dataset (Krizhevsky, 2009) can be found in `https://www.cs.toronto.edu/~kriz/cifar.html`. Code for the ResNet20 model can be found in `https://github.com/google-research/google-research/blob/master/cold_posterior_bnn/models.py` under the Apache License, Version 2.0.

### G.2  ERROR VARIANCE OF THE GAUSSIAN REGRESSION MODEL

As the training data $y$ in our experiment is standardized to unit variance, it is safe to assume that error variance $\sigma^2$ of the trained model will be less than 1. Therefore, we set $\sigma^2 = 0.1$.

### G.3  PRIOR VARIANCE

We use an isotropic Gaussian $\mathcal{N}(0, \sigma_p^2)$ for the neural network weights. The variance $\sigma_p^2$ is specified in Table 4.

# H  SGMCMC

We use the implementation of SGMCMC as presented in Wenzel et al. (2020). This corresponds to the stochastic gradient Hamiltonian Monte Carlo (Chen et al., 2014) with rescaled hyperparameters and an adaptive scaling on the Gaussian noise to ensure efficient sampling from a tempered posterior. The samples of $p_\beta(\theta|\mathcal{D}) \propto \exp\{-\beta U(\theta)\} = \exp\{-U(\theta)/T\}$, where $U(\cdot)$ is the *posterior energy function* defined as

$$U(\theta) := -\log p(\theta) - \sum_{x,y \in \mathcal{D}} \log p(y|x, \theta), \tag{10}$$

can be drawn by simulating the following Langevin stochastic difference equation (SDE) over $\theta \in \mathbb{R}^d$ and momentum $m \in \mathbb{R}^d$

$$\mathrm{d}\theta = \mathbf{M}^{-1} m \, \mathrm{d}t, \tag{11}$$

$$\mathrm{d}m = -\nabla_\theta U(\theta) \, \mathrm{d}t - \gamma m \, \mathrm{d}t + \sqrt{2\gamma T} \mathbf{M}^{1/2} \mathrm{d}\mathbf{W}, \tag{12}$$

for any *friction* $\gamma > 0$. Here, $\mathbf{W}$ is a Wiener process, which can be loosely interpreted as a generalized Gaussian distribution (Leimkuhler & Matthews, 2015). The *mass matrix* $\mathbf{M}$ is a preconditioner that can help in speeding up the convergence to the limiting distribution of this SDE. We also prefer working with $T$ instead of the inverse temperature $\beta$ in the sampler to facilitate sampler diagnostics and temperature ramp-up.

In practice, the gradient of $U(\theta)$ is approximated by a minibatch gradient estimator

$$\nabla_\theta \tilde{U}(\theta) := -\nabla_\theta \log p(\theta) - \frac{|\mathcal{D}|}{|\mathcal{B}|} \sum_{x,y \in \mathcal{B}} \nabla_\theta \log p(y|x, \theta),$$

where $\mathcal{B}$ denotes a minibatch, and $|\mathcal{B}|$ and $|\mathcal{D}|$ denote the batch size and number of training samples respectively. The SDEs are then solved numerically with a first-order symplectic Euler discretization scheme using this minibatch gradient estimator, resulting in the following update equations

$$m^{(t)} = (1 - h\gamma)m^{(t-1)} - h\nabla_\theta \tilde{U}(\theta^{(t-1)}) + \sqrt{2\gamma h T} \mathbf{M}^{1/2} \mathbf{R}^{(t)} \tag{13}$$

$$\theta^{(t)} = \theta^{(t-1)} + h\mathbf{M}^{-1} m^{(t)}$$

where $\mathbf{R}^{(t)} \sim \mathcal{N}(0, I_d)$ is a standard Gaussian vector. Note that the temperature shows up in the update equations and effectively scales the random noise. This is helpful for drawing samples from cold posteriors, which tend to have narrow, high density regions. The scaling prevents the Markov chain from taking steps that are too large and missing the high-density region.

The step size $h$ is often also modulated with a scheduler $C(t) : \mathbb{R}^+ \to [0, 1]$

$$h = h_0 C(t), \tag{14}$$

where $h_0$ is the initial step size. Where appropriate, we also use the layerwise preconditioner as proposed in Wenzel et al. (2020) to speed up convergence and reduce approximation error.

# I  SGMCMC DIAGNOSTICS

In this work, we use the kinectic temperature diagnostic (Wenzel et al., 2020) to assess the quality of SGMCMC samples. This is a departure from the common practice of using test LPD as a proxy for the posterior approximation error, since this been shown to be a poor proxy (Deshpande et al., 2022). In addition, we also report the rank-normalized split-$\hat{R}$ statistics (Vehtari et al., 2021) on the (unnormalized) posterior density, which are invariant to the permutation of neural network weights.

## I.1  KINETIC TEMPERATURE

We report the expected kinetic temperature of each SGMCMC chain at different temperatures, as proposed in Wenzel et al. (2020, Appendix I). The kinetic temperature estimator is given by

$$\hat{T}(m) = \frac{m^\top \mathbf{M}^{-1} m}{\dim(m)} \tag{15}$$

where $m$ and $\mathbf{M}$ are the (random) momentum and the mass matrix in the SDE (11)-(12). For a perfect simulation of the SDE, (15) is an unbiased estimator of the temperature of the system, i.e., $\mathbb{E}[\hat{T}(m)] = T$ (Leimkuhler & Matthews, 2015, Section 6.1.5).

In Table 5, the expected kinetic temperatures were computed by averaging over the temperature samples over the whole Markov chain. We use these estimates to gauge the simulation quality of the SDEs — an estimate closer to the target temperature indicates a better approximation of the SDEs. We generally expect the simulation quality to worsen in the lower temperature regime, due to the errors from both discretization and gradient sub-sampling becoming more prominent relative to the randomness in $m$. Otherwise, there is no major concern about the sample quality, as observed in Table 5.

## I.2 Rank-normalized split-$\hat{R}$

In Table 6, we report the rank-normalized, split-$\hat{R}$ (Vehtari et al., 2021) on the potential energy (10), which is invariant to the permutation of neural network weights. This statistic is a strict improvement over the traditional potential scale reduction factor $\hat{R}$ (Gelman & Rubin, 1992) and split-$\hat{R}$ (Gelman et al., 2013). The $\hat{R}$ statistic compares between-chain and within-chain variances, thus requiring multiple independently-initialized Markov chains to compute. It was designed based on the idea that, for Markov chains that are mixing well, they should converge to the same limiting distribution regardless of initialization. Therefore, the between-chain and within-chain variances should roughly be the same, and a $\hat{R}$ closer to 1 is considered better. The term 'split' indicates that the Markov chains are divided in half, resulting in double the number of Markov chains with half the original length. The variances are computed across these doubled number of Markov chains to detect poor convergence within an original chain. The 'traditional' $\hat{R}$ is computed on the original values of the potential energy, while the 'rank-normalized' $\hat{R}$ is computed on the rank-normalized values of the potential energy and does not require the limiting distribution to have a finite mean or variance.

In Bayesian deep learning, the posterior distribution is almost always multi-modal (Izmailov et al., 2021), and we should not expect the independently-initialized Markov chains to always converge to the same limiting distribution. Therefore, we should not be overly alarmed by the relatively 'large' readings of $\hat{R}$ in Table 6. These values are large by the usual standard in Bayesian statistics but are common in Bayesian deep learning (Izmailov et al., 2021; Fortuin et al., 2022).

## J Hyperparameters and the Compute Environment

In this section, we provide a detailed explanation of the hyperparameters for SGD and SGMCMC. The values are summarized in Table 7.

**Learning rate and scheduler (SGD).** This is the learning rate scheduler. The cosine schedule starts from the indicated learning rate and gradually decreases to 0 throughout the entire SGD run. The 'piecewise' scheduler is used in Wenzel et al. (2020), where the initial learning rate is multiplied by a value at specified epochs. We write it in the format of (epoch, multiplier): (80, 0.1), (120, 0.01), (160, 0.001), (180, 0.0005).

**Weight decay.** We subtract a weight decay term from the transformed gradient before scaling the transformed gradient with a learning rate. This subtraction is performed because we are maximizing the likelihood rather than minimizing a loss.

**Gradient clipping.** We clip the gradient norm to a specified threshold, as implemented in Optax (DeepMind et al., 2020). This helps stabilize the training procedure.

**Learning rate and momentum (SGMCMC).** The learning rate and momentum terms will control both the initial step size $h_0$ and the friction $\gamma$. The relationship between them is given in Wenzel et al. (2020, Appendix B) and is repeated here:

$$h_0 = \sqrt{learning\ rate/n}, \quad \gamma = (1 - momentum)/h_0,$$

where $n$ is the size of the training set.

**Scheduler and cycle length (SGMCMC).**   This is the scheduler $C(t)$ in (14) that modulates $h$. During the burn-in phase, the scheduler is fixed at 1, i.e. $h = h_0$. Then, during the sampling phase, the step size is modulated with a cyclical cosine schedule (Zhang et al., 2020) with a period of the specified epoch. One sample is collected at the end of each cycle.

**Ramp start and end.**   As we run our SGMCMC algorithm from $T = 0$ and only start increasing $T$ after a specified number of epochs, these two hyperparameters indicate the epoch when $T$ was gradually increased and the epoch when $T$ reaches the target temperature.

**Burn-in epochs, total epochs and usable samples (SGMCMC).**   The 'total epochs' indicate the total number of epochs run by SGMCMC, including epochs during the burn-in phase. 'Usable samples' indicate the number of samples collected after the burn-in phase.

### J.1   Computational environment and resources

The algorithms were implemented in `JAX` (Bradbury et al., 2018). We used `Flax` (Heek et al., 2023) and `Tensorflow Probabilities` (Dillon et al., 2017) to implement neural networks and models. Plots were generated with `tidyverse` (Wickham et al., 2019) and `ggplot2` (Wickham, 2016). Rank-normalized split-$\hat{R}$ was computed with the `posterior` package (Bürkner et al., 2023). We also used `xarray` (Hoyer & Hamman, 2017) and `arviz` (Kumar et al., 2019) to conduct explanatory analysis, and `wandb` (Biewald, 2020) to track our experiments. Our code will be released.

The experiment was conducted on Google Cloud Platform utilizing TPU-V3. In each machine, there are 8 TPU cores with 16GB of TPU memory attached to each core (i.e., totaling 128GB of memory in each machine). However, we only utilize one TPU core due to the difficulty in parallelizing the MCMC chains across different cores, and effectively only have access to 16GB of memory in each machine.

On a CIFAR10 experiment, a SGD run (for our proposed algorithm) takes roughly 0.5 hour to complete. A SGMCMC run at a particular temperature takes roughly 4.5 hours. In our main experiment, as we are computing over a grid of 9 temperature plus the optimal temperature, one repetition of the experiment takes 45.5 hours. Multiplying this by five repetitions and we get 227.5 hours to produce one of the CIFAR10 subplot in Figure 1. The rest of the subplot are considerably cheaper: 16 hours for MNIST, and less than an hour for each of Concrete, Naval and Energy.

In total, it takes 474 hours to generate the posterior samples for computing Table 1 and Figure 1. It takes an additional 279 hours for the experiment that varies data augmentation strength, i.e., Figure 2. Hyperparamters tuning takes an additional 200 hours. We have also spent roughly 100 compute-hours to try out various prior for $\beta$.

### J.2   Standard error calculation

The standard error (SE) of the means are computed with `sd` routine in `R`. More specifically, it computes the square root of an unbiased estimate of the variance. The upper and lower bound of the shaded areas in all figures are 'mean $+$ SE' and 'mean $-$ SE' respectively.

## K   Extra Figures and Tables From the Main Experiment

In this section, we provide additional figures and tables to complement the results presented in the main text.

We first show the validation LPD plotted against the temperature in Figure 3 and find that our method can recover temperatures in the high-performing regions. We further improve the visualization of Figure 1 by separating the test LPD computed with SM-PD (3) and TM-PD (8) into Figure 4 and Figure 5, respectively. A tabular version of the results is also provided in Table 8 and Table 9, with an SGD reference included in all the results.

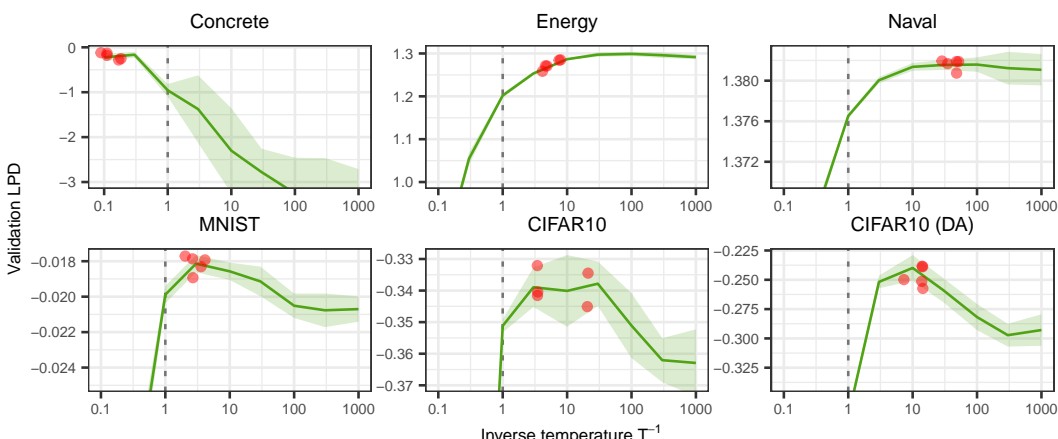

Figure 3: Validation LPD plotted against inverse temperature $\beta$. This is computed with SM-PD as defined in (3). Solid lines and shaded area represent mean $\pm$ standard error across five repetitions. The vertical dotted lines indicate the PPD at $\beta = 1$. There are five red dots in each plot, each of them corresponding to a repetition of $\hat{\beta}^*$ from our method. Higher LPD indicates better performance.

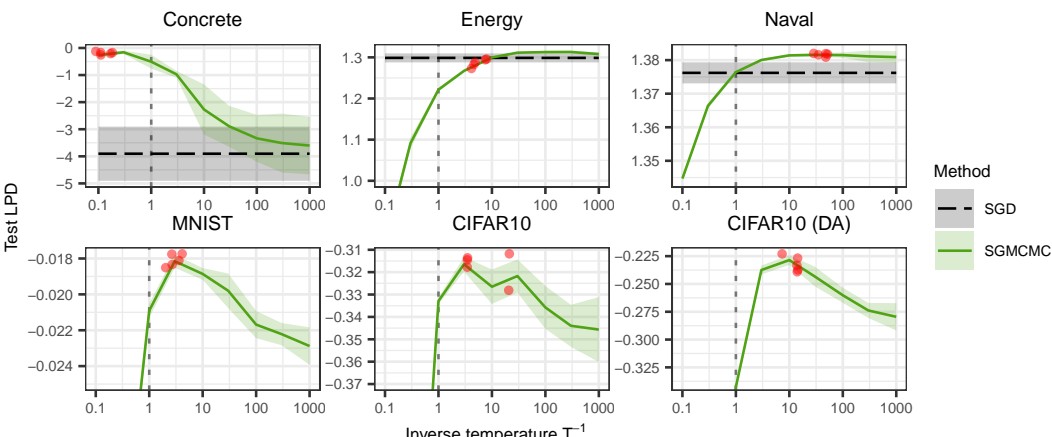

Figure 4: Test LPD plotted against inverse temperature $\beta$ with SGMCMC (green, solid). This is computed with SM-PD as defined in (3). The SGD solution (horizontal, black, dashed) is included as a reference. The SGD reference in MNIST and CIFAR10 examples performs considerably worse and is out of range. Lines and shaded area represent mean $\pm$ standard error across five repetitions. The vertical dotted lines indicate the PPD at $\beta = 1$. There are five red dots in each plot, each of them corresponding to a repetition of $\hat{\beta}^*$ from our method. Higher LPD indicates better performance.

Finally, we show the performance of point predictions across different temperatures in Figure 6, complementing Table 1. Note that the point predictions are identical for SM-PD and TM-PD. We observe that predictions from tempered posteriors generally outperform SGD.

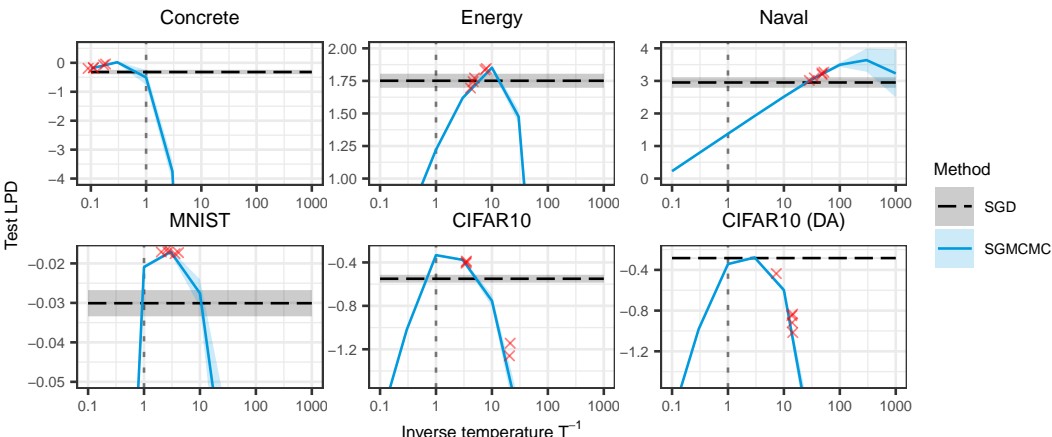

Figure 5: Test LPD plotted against inverse temperature $\beta$ with SGMCMC (blue, solid). This is computed with TM-PD as defined in (8). The SGD solution (horizontal, black, dashed) is included as a reference. Lines and shaded area represent mean $\pm$ standard error across five repetitions. The vertical dotted lines indicate the PPD at $\beta = 1$. There are five red dots in each plot, each of them corresponding to a repetition of $\hat{\beta}^*$ from our method. Higher LPD indicates better performance.

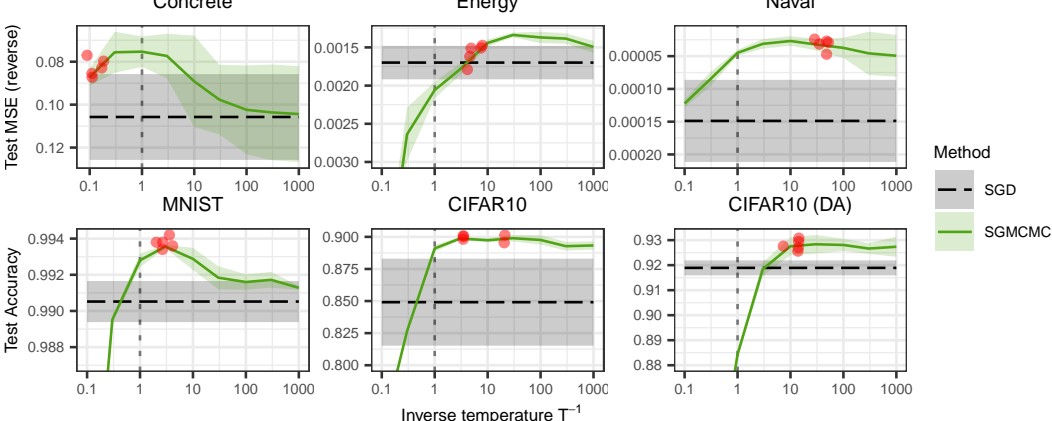

Figure 6: Test MSE (Concrete, Energy, Naval) and accuracy (MNIST, CIFAR10) of the point predictions of the PPDs, plotted against inverse temperature $\beta$. The results for SM-PD (3) and TM-PD (8) are consolidated, as both produce identical point predictions by definition. The SGD solution (horizontal, black, dashed) is included as a reference. Lines and shaded area represent mean $\pm$ standard error across five repetitions. The vertical dotted lines indicate the PPD at $\beta = 1$. There are five red dots in each plot, each of them corresponding to a repetition of $\hat{\beta}^*$ from our method. Note the MSE has been reversed, and higher value indicates better performance in all plots.

## L  THE DECISION-THEORETIC GENERALIZED BAYESIAN FRAMEWORK

The classical Bayesian rule (1) updating prior belief $p(\theta)$ to the posterior belief $p(\theta|\mathcal{D})$ arises from Bayes' Theorem. An implicit assumption in this procedure for optimal performance is that the model is correctly specified: there exists some $\theta_0$ such that $p(y|x, \theta_0) = q(y|x)$ for all $(x, y)$. The adage that "all models are wrong" (Box, 1976) highlights that this assumption should not be expected to hold. Consequently, it does not make sense to infer on a parameter $\theta$ that lacks a connection to the true data-generation distribution.

Using a decision-theoretic argument, Bissiri et al. (2016) offers an alternative justification for using (1), even when the model is misspecified. Suppose that we are interested in a quantity $\theta_0$ that minimizes the expected loss under the truth $q(x, y)$, that is,

$$\theta_0 := \arg\min_\theta \int \ell(\theta, x, y) q(x, y) \, \mathrm{d}y \, \mathrm{d}x \tag{16}$$

for some loss function $\ell$. Then, we would like to derive an update rule $\psi$ that takes an observation $(x_1, y_1)$ and updates our prior belief on $\theta_0$ to a posterior belief,

$$p(\theta|x_1, y_1) = \psi\{\ell(\theta, x_1, y_1), p(\theta)\}.$$

Furthermore, the update rule should be *coherent*. That is, suppose that we would like to update our posterior with two data points $\{(x_1, y_1), (x_2, y_2)\}$, the update rule should satisfy

$$\psi\{\ell(\theta, x_2, y_2), \psi\{\ell(\theta, x_1, y_1), p(\theta)\}\} \equiv \psi\{\ell(\theta, x_2, y_2) + \ell(\theta, x_1, y_1), p(\theta)\}.$$

The coherent property will ensure that the same posterior arises regardless of the order which the data are processed. Bissiri et al. (2016) shows that all coherent update rules take the form

$$p_{\mathrm{GB}}(\theta|\mathcal{D}) \propto p(\theta) \exp\left( -\sum_{(x,y)\in\mathcal{D}} \ell(\theta, x, y) \right), \tag{17}$$

which we will refer to as the *general Bayesian update*, and the associated posterior $p_{\mathrm{GB}}(\theta|\mathcal{D})$ as the *generalized Bayes posterior*, which has been widely analyzed (Zhang, 2006a;b; Jiang & Tanner, 2008; Bhattacharya et al., 2019; Alquier et al., 2016; Alquier & Ridgway, 2020). Note that we must take care to have a finite normalizing constant to (17),

$$0 < \int \exp\left( -\sum_{(x,y)\in\mathcal{D}} \ell(\theta, x, y) \right) p(\theta)\mathrm{d}\theta < +\infty.$$

Under certain regularity conditions, the generalized Bayes posterior will concentrate on $\theta_0$ as $n \to \infty$; a proof can be found in McLatchie et al. (2024, Lemma 5). The key advantage of this framework in lies in its ability to directly infer the quantity of interest (i.e., $\theta_0$) using a loss function, without necessitating any probabilistic assumptions that relate $\theta_0$ to the data. For formal proofs, we direct interested readers to (Bissiri et al., 2016, Section 1).

### L.1  CALIBRATING INFORMATION GAIN

The tempered posterior, as defined in (2), is a special parameterisation of the generalized Bayes posterior achieved by setting the loss to be $-\log p(y|x, \theta)$ and scaling it by a factor of $\beta > 0$, i.e. $\ell(\theta, x, y) = -\beta \log p(y|x, \theta)$, and concentrating the prior to an equivalent extent. Therefore, the tempered posterior has an interpretation of a coherent update rule for the KL minimizer $\theta_\dagger = \arg\min_\theta -\beta \log p(y|x, \theta) = \arg\min_\theta D_{\mathrm{KL}}(q(y|x)\|p(y|x, \theta))$, as defined in the main text. From this perspective, the temperature has the role of controlling the amount of information being 'added' to the prior in each update step. Holmes & Walker (2017) proposed setting $\beta$ such that the *expected information gain* of our setup (LHS of (18)) matches that of a hypothetical experiment (RHS of (18)),

$$\int D(p_\beta(\theta|\mathcal{D}), p(\theta)) q(x, y) \mathrm{d}y\mathrm{d}x = \int D(p(\theta|\mathcal{D}), p(\theta)) p(y|x, \theta_\dagger) q(x) \mathrm{d}y\mathrm{d}x \tag{18}$$

where $D$ is a divergence that measures the information gain from a prior to a posterior, and $p_\beta$ is a tempered posterior (with likelihood tempering only).

On the LHS of (18), we are measuring the expected information gain in our existing setup, i.e., $q(x, y)$ is unknown, from a prior to a tempered posterior $p_\beta$. Note that this integral is a function of $\beta$. Then, Holmes & Walker (2017) proposed picking a $\beta$ that matches up the expected information gain from a well-specified experiment that targets the same quantity of interest $\theta_\dagger$, i.e., having the truth be $p(\cdot|x, \theta_\dagger)q(x)$. In the well-specified setting, the optimal $\beta$ is 1, and thus the information gain is computed from the prior to a posterior at $\beta = 1$. This proposed method is based on the principle that the expected information gain should match across posteriors that are targeting the same quantity of interest. In their follow up work, Lyddon et al. (2019) proposed selecting $\beta$ such that the asymptotic Fisher information number of the generalized Bayes posterior to match to that derived from the "loss-likelihood" bootstrap method, which is a sampling method for general Bayes posteriors.

### L.2 Data augmentation in generalized Bayes

Accounting for data augmentation in BDL is non-trivial, as the data are no longer i.i.d. This has led to the development of several techniques to address this issue (Nabarro et al., 2022; Kapoor et al., 2022). However, data augmentation can be easily incorporated into decision-theoretic GB (Bissiri et al., 2016), as the generalized Bayes posterior is derived from iterative belief updates rather than relying on the independence assumption. We can then regard the posterior as reflecting our belief in the KL minimizer from the 'augmented' truth to our model.

For example, in the ResNet20-CIFAR10 example with SGMCMC in Wenzel et al. (2020), the data was augmented in each epoch, and we effectively have $Mn$ (instead of $n$) number of observations in the training set $\tilde{\mathcal{D}} = \{(\tilde{x}_i, y_i)\}_{i=1}^{Mn}$, where $M$ is the number of epochs, and $\tilde{x}_i$ represents an augmented image. Then, setting $\ell(x, y, \theta) = -\log p(y|x, \theta)$ and working 'backward', the posterior can be regarded as representing our belief in a new quantity $\tilde{\theta}_\dagger$. This quantity is the minimizer of the KL divergence from an 'augmented' truth $\tilde{q}(x, y) = \tilde{q}(y|x)\tilde{q}(x)$, from which the augmented data have arisen, to the model $p(y|x, \theta)$

$$\tilde{\theta}_\dagger = \arg\min_\theta \mathbb{E}_{\tilde{q}(x)} D_{\mathrm{KL}}(\tilde{q}(y|x)\|p(y|x, \theta)).$$

Table 5: The expected kinetic temperatures of each SGMCMC chain are presented here. In this table, the temperatures are inverted. Kinetic temperatures that are closer to the target indicate better simulation of the SDE. The kinetic temperatures in the $\hat{\beta}^*$ row differ across repetitions due to the variety of $\hat{\beta}^*$ obtained from SGD in each repetition.

| TARGET $\beta$ | | CONCRETE | ENERGY | NAVAL | MNIST | CIFAR10 | CIFAR10 (DA) |
|---|---|---|---|---|---|---|---|
| 0.1 | REP. 1 | 0.08 | 0.08 | 0.10 | 0.10 | 0.10 | 0.10 |
| | REP. 2 | 0.10 | 0.09 | 0.10 | 0.10 | 0.10 | 0.10 |
| | REP. 3 | 0.09 | 0.09 | 0.10 | 0.10 | 0.10 | 0.10 |
| | REP. 4 | 0.08 | 0.09 | 0.10 | 0.10 | 0.10 | 0.10 |
| | REP. 5 | 0.09 | 0.09 | 0.10 | 0.10 | 0.10 | 0.10 |
| 0.3 | REP. 1 | 0.29 | 0.24 | 0.30 | 0.30 | 0.30 | 0.30 |
| | REP. 2 | 0.30 | 0.29 | 0.30 | 0.30 | 0.30 | 0.30 |
| | REP. 3 | 0.30 | 0.30 | 0.30 | 0.30 | 0.30 | 0.30 |
| | REP. 4 | 0.30 | 0.26 | 0.30 | 0.30 | 0.30 | 0.30 |
| | REP. 5 | 0.30 | 0.25 | 0.30 | 0.30 | 0.30 | 0.30 |
| 1 | REP. 1 | 1.00 | 1.00 | 1.00 | 1.00 | 1.00 | 1.00 |
| | REP. 2 | 0.99 | 1.00 | 0.98 | 1.00 | 1.00 | 1.00 |
| | REP. 3 | 0.99 | 0.99 | 1.00 | 1.00 | 1.00 | 1.00 |
| | REP. 4 | 0.99 | 1.00 | 0.99 | 1.00 | 1.00 | 1.00 |
| | REP. 5 | 1.00 | 0.99 | 0.99 | 1.00 | 1.00 | 1.00 |
| 3 | REP. 1 | 2.98 | 2.97 | 2.98 | 3.00 | 2.99 | 2.99 |
| | REP. 2 | 2.95 | 3.00 | 2.95 | 3.00 | 2.99 | 2.99 |
| | REP. 3 | 2.98 | 2.99 | 2.98 | 3.00 | 2.99 | 2.99 |
| | REP. 4 | 2.98 | 2.99 | 2.95 | 3.00 | 2.99 | 2.99 |
| | REP. 5 | 2.96 | 2.98 | 2.96 | 3.00 | 2.99 | 2.99 |
| 10 | REP. 1 | 9.87 | 9.94 | 9.82 | 9.99 | 9.96 | 9.92 |
| | REP. 2 | 9.88 | 10.00 | 9.69 | 9.99 | 9.96 | 9.93 |
| | REP. 3 | 9.87 | 9.97 | 9.89 | 9.99 | 9.96 | 9.92 |
| | REP. 4 | 9.88 | 9.97 | 9.68 | 9.99 | 9.96 | 9.93 |
| | REP. 5 | 9.90 | 9.97 | 9.78 | 9.99 | 9.96 | 9.93 |
| 30 | REP. 1 | 29.61 | 29.80 | 28.48 | 29.98 | 29.77 | 29.50 |
| | REP. 2 | 29.28 | 29.85 | 28.64 | 29.98 | 29.77 | 29.59 |
| | REP. 3 | 29.39 | 29.76 | 28.73 | 29.96 | 29.77 | 29.54 |
| | REP. 4 | 28.93 | 29.91 | 27.61 | 29.96 | 29.77 | 29.55 |
| | REP. 5 | 29.04 | 29.85 | 28.73 | 29.97 | 29.77 | 29.63 |
| 100 | REP. 1 | 95.81 | 99.07 | 84.76 | 99.87 | 98.21 | 96.34 |
| | REP. 2 | 92.08 | 98.88 | 85.24 | 99.88 | 98.20 | 97.12 |
| | REP. 3 | 92.25 | 97.94 | 85.66 | 99.83 | 98.19 | 96.58 |
| | REP. 4 | 92.20 | 99.26 | 72.74 | 99.81 | 98.15 | 96.69 |
| | REP. 5 | 91.28 | 98.51 | 87.84 | 99.86 | 98.21 | 97.34 |
| 300 | REP. 1 | 259.87 | 293.28 | 196.04 | 299.19 | 285.58 | 276.00 |
| | REP. 2 | 248.86 | 293.13 | 207.08 | 299.19 | 285.32 | 280.62 |
| | REP. 3 | 228.33 | 287.56 | 187.88 | 299.06 | 285.15 | 276.42 |
| | REP. 4 | 221.00 | 289.86 | 133.78 | 298.99 | 285.40 | 276.79 |
| | REP. 5 | 229.55 | 293.08 | 213.83 | 299.14 | 285.24 | 281.91 |
| 1000 | REP. 1 | 547.10 | 939.37 | 346.19 | 992.38 | 860.32 | 775.48 |
| | REP. 2 | 649.51 | 943.01 | 399.87 | 992.25 | 857.79 | 821.73 |
| | REP. 3 | 503.20 | 875.31 | 286.99 | 991.97 | 853.45 | 792.92 |
| | REP. 4 | 436.40 | 880.59 | 202.85 | 991.61 | 853.14 | 798.56 |
| | REP. 5 | 446.14 | 958.23 | 432.83 | 991.98 | 855.56 | 844.81 |
| $\hat{\beta}^*$ | REP. 1 | 0.16 | 7.44 | 47.53 | 2.65 | 21.04 | 7.28 |
| | REP. 2 | 0.17 | 4.86 | 44.12 | 2.68 | 3.49 | 13.93 |
| | REP. 3 | 0.11 | 7.91 | 33.33 | 4.13 | 3.44 | 13.72 |
| | REP. 4 | 0.08 | 4.12 | 42.88 | 3.58 | 3.46 | 14.17 |
| | REP. 5 | 0.10 | 4.58 | 27.26 | 2.03 | 20.48 | 14.28 |

Table 6: The split-$\hat{R}$ values for the log-posterior at different (inverse) temperatures. These split-$\hat{R}$ values are computed from 5 SGMCMC chains initialized at different $\theta$. Values closer to 1 indicate better mixing of the Markov chains. $\hat{\beta}^*$ is left out as the temperature (and hence the posterior) differs across repetitions.

| $\beta$ | CONCRETE | ENERGY | NAVAL | MNIST | CIFAR10 | CIFAR10 (DA) |
|---|---|---|---|---|---|---|
| 0.1 | 1.26 | 1.19 | 1.82 | 1.07 | 1.01 | 1.04 |
| 0.3 | 1.04 | 1.16 | 1.83 | 1.03 | 1.02 | 1.08 |
| 1 | 1.54 | 1.09 | 1.94 | 1.02 | 1.13 | 1.07 |
| 3 | 2.09 | 1.10 | 2.24 | 1.03 | 1.17 | 1.46 |
| 10 | 2.79 | 1.30 | 2.78 | 1.14 | 1.69 | 1.99 |
| 30 | 2.68 | 1.77 | 3.07 | 1.27 | 1.42 | 2.30 |
| 100 | 2.83 | 2.38 | 3.25 | 1.27 | 1.24 | 2.29 |
| 300 | 2.81 | 2.40 | 3.10 | 1.19 | 1.19 | 2.19 |
| 1000 | 3.97 | 2.87 | 3.20 | 1.16 | 1.15 | 2.38 |

Table 7: Hyperparameters for SGD and SGMCMC. We use the same set of hyperparameters for both CIFAR-10 experiments, with or without data augmentation.

| | | CONCRETE | ENERGY | NAVAL | MNIST | CIFAR10 |
|---|---|---|---|---|---|---|
| SGD | LEARNING RATE | $10^{-6}$ | $10^{-6}$ | $10^{-8}$ | $10^{-6}$ | $10^{-6}$ |
| | SCHEDULER | COSINE | COSINE | COSINE | COSINE | PIECEWISE |
| | MOMENTUM | 0.9 | 0.9 | 0.9 | 0.9 | 0.9 |
| | NESTEROV | NO | NO | NO | NO | NO |
| | WEIGHT DECAY | 1 | 1 | 1 | 1 | 500 |
| | BATCH SIZE | FULL | FULL | 128 | 128 | 128 |
| | TOTAL EPOCHS | 15000 | 15000 | 10000 | 10 | 200 |
| | GRADIENT CLIPPING | $10^6$ | $10^6$ | $10^4$ | $10^6$ | $10^6$ |
| SGMCMC | LEARNING RATE | $10^{-3}$ | $10^{-3}$ | $10^{-4}$ | 0.01 | 0.1 |
| | SCHEDULER | CYCLICAL | CYCLICAL | CYCLICAL | CYCLICAL | CYCLICAL |
| | CYCLE LENGTH | 200 | 200 | 100 | 10 | 50 |
| | PRECONDITIONER | NONE | NONE | NONE | NONE | LAYERWISE |
| | MOMENTUM | 0.98 | 0.98 | 0.98 | 0.98 | 0.98 |
| | BATCH SIZE | FULL | FULL | 128 | 128 | 128 |
| | RAMP START (EPOCH) | 4800 | 4800 | 900 | 10 | 100 |
| | RAMP END (EPOCH) | 5000 | 5000 | 1000 | 20 | 150 |
| | BURN-IN EPOCHS | 10000 | 10000 | 5000 | 200 | 500 |
| | TOTAL EPOCHS | 30000 | 30000 | 15000 | 1200 | 2000 |
| | USABLE SAMPLES | 100 | 100 | 100 | 100 | 30 |
| | GRADIENT CLIPPING | $10^6$ | $10^6$ | $10^6$ | $10^6$ | $10^6$ |

Table 8: Test LPD of the regression models. The values presented are means $\pm$ standard errors across five repetitions, with the best value among the four methods highlighted in bold. TM-PDs generally have better performance for the regression models. Higher LPD indicates better performance.

| | METHOD | CONCRETE | ENERGY | NAVAL |
|---|---|---|---|---|
| SM-PD | SGD | $-3.904 \pm 1.000$ | $1.299 \pm 0.011$ | $1.376 \pm 0.003$ |
| | $\beta = 1$ | $-0.506 \pm 0.255$ | $1.221 \pm 0.005$ | $1.376 \pm 0.000$ |
| | $\beta = \hat{\beta}^*$ | $\mathbf{-0.186 \pm 0.049}$ | $1.287 \pm 0.009$ | $\mathbf{1.382 \pm 0.000}$ |
| | GRID | $-0.216 \pm 0.121$ | $\mathbf{1.312 \pm 0.003}$ | $\mathbf{1.382 \pm 0.001}$ |
| TM-PD | SGD | $-0.319 \pm 0.076$ | $1.75 \pm 0.054$ | $2.95 \pm 0.162$ |
| | $\beta = 1$ | $-0.506 \pm 0.255$ | $1.22 \pm 0.005$ | $1.38 \pm 0.000$ |
| | $\beta = \hat{\beta}^*$ | $-0.13 \pm 0.069$ | $1.78 \pm 0.063$ | $3.16 \pm 0.106$ |
| | GRID | $\mathbf{0.019 \pm 0.035}$ | $\mathbf{1.85 \pm 0.017}$ | $\mathbf{3.71 \pm 0.194}$ |

Table 9: Test LPD of the classification models. The presented values are the means $\pm$ standard error across five repetitions, with the best value among the four methods boldfaced. SM-PDs tend to perform better on CIFAR-10, while SM-PD and TM-PD have similar performance on MNIST. Higher LPD indicates better performance.

| | METHOD | MNIST | CIFAR10 | CIFAR10 (DA) |
|---|---|---|---|---|
| SM-PD | SGD | $-0.076 \pm 0.032$ | $-1.138 \pm 0.306$ | $-1.234 \pm 0.176$ |
| | $\beta = 1$ | $-0.021 \pm 0.000$ | $-0.333 \pm 0.001$ | $-0.343 \pm 0.002$ |
| | $\beta = \hat{\beta}^*$ | $\mathbf{-0.018 \pm 0.000}$ | $\mathbf{-0.317 \pm 0.006}$ | $-0.232 \pm 0.007$ |
| | GRID | $\mathbf{-0.018 \pm 0.000}$ | $-0.319 \pm 0.003$ | $\mathbf{-0.229 \pm 0.004}$ |
| TM-PD | SGD | $-0.030 \pm 0.003$ | $-0.55 \pm 0.036$ | $-0.284 \pm 0.004$ |
| | $\beta = 1$ | $-0.021 \pm 0.000$ | $\mathbf{-0.333 \pm 0.001}$ | $-0.343 \pm 0.002$ |
| | $\beta = \hat{\beta}^*$ | $\mathbf{-0.017 \pm 0.000}$ | $-0.719 \pm 0.443$ | $-0.811 \pm 0.221$ |
| | GRID | $\mathbf{-0.017 \pm 0.001}$ | $\mathbf{-0.333 \pm 0.001}$ | $-0.279 \pm 0.007$ |

