# OpenReview forum: "Temperature Optimization for Bayesian Deep Learning"
_ICLR.cc/2025/Conference — Submitted to ICLR 2025_

### Official Review · Reviewer_UMPW · 2024-10-28

**Soundness:** 3
**Presentation:** 3
**Contribution:** 2
**Rating:** 3
**Confidence:** 3

**Summary:**

This work  introduces a method for training the temperature parameter $\beta$ unlike the existing approach for Cold Posterior Effect (CPE), where the temperature parameter is set as the hyperparameter. To this end, the authors propose optimizing the posterior predictive density $p_{\beta}(y|x,\mathcal{D},\theta)$ (SM-PD) or the log likelihood of the tempered model $p(y|x,\theta,\beta)$ (TM-PD), directly and updating the model parameters $\theta$ and temperature parameter $\beta$ jointly. For regression task, the authors provide a rationale explaining how the proposed objective enables learning of the temperature parameter $\beta$. Empirically, they also demonstrate that the training the temperature parameter can yield the same performance with the performance of the grid search on UCI regression task and CIFAR 10 classification task.

**Strengths:**

* This work tackles an interesting question for CPE that trains the temperature parameter while preserving the CPE.

* Training the temperature parameters seems marginally effective for training with domain augmentation.

* The related works are described well.

**Weaknesses:**

* The proposed methods seems less consistent according to the task type because the TM-PD is effective for regression and the SM-PD  is effective for classification in obtaining the optimal temperature parameter.

* The advantage of learning the temperature parameter are not clear because the performances of the trained temperature in Table 1 seem similar to those of the grid search.

**Questions:**

* Do you think that why the effective learning objective for the optimal temperature parameter is different depending on the task type ? Is there any way to make consistency of the learning objective in regardless of the task type ?


*  As mentioned above, the advantage of learning the temperature parameter does not seem clear based on the given manuscript. Are there other advantages such as computational efficiency or ease of implementation? If they exist, presenting those advantages would be better to persuade the necessity of learning the temperature parameters.

*  Based on Figure 6 in [1], the CPE still holds for varying batch size. However, the proposed methods have not been demonstrated for varying batch size. In the setting of Table 1, would the SM-PD and TM-PD still be effective in finding the consistent optimal temperature parameter if they are applied over varying batch sizes $\\{32,64,128,256\\}$?

[1] How Good is the Bayes Posterior in Deep Neural Networks Really? - ICML 20

---

### Official Review · Reviewer_ty2k · 2024-11-02

**Soundness:** 3
**Presentation:** 3
**Contribution:** 2
**Rating:** 5
**Confidence:** 4

**Summary:**

The paper discusses the socalled cold posterior effect in Bayesian deep learning, and presents a data driven method to estimate an optimal temperature. The method is demonstrated on standard regression and classification benchmarks. The includes detailed theoretical discussions surrounding the topic of the cold posterior effect.

**Strengths:**

The paper is notationally clear and seems solid.

The topic is important and of interest to the community.

Theories related to the cold posterior effect are discussed in good detail.

Additional material in the appendix is illuminating.

Source code is published for reproducibity

**Weaknesses:**

My primary critique is that the empirical study is somewhat limited. While the proposed method appears straightforward and effective, its comparative performance with alternative approaches is not entirely clear. For instance, while the paper demonstrates how the tempered posterior can be reframed as an equivalent 'tempered model,' it does not include a comparison with direct Bayesian inference within this model, treating the temperature as a model parameter. Another common technique involves fitting the temperature posthoc on validation data, which is also absent from the analysis. Overall, the paper leaves it unclear how this method distinguishes itself from other approaches. Additionally, given that the proposed method essentially follows a specific two-step procedure for inference in models with certain parameterizations, there are numerous other inference procedures that could be relevant for comparison.

**Questions:**

The introduction seems to identify BDL with sampling. Could it be stated clearly which types of approximate bayesian inference this work applies to?

"Our primary interest in this work..." The abstract alludes to other important metrics, such as calibration and robustness. Why is the focus here on LPD?

You state "warmer temperatures are often better ... only for metrics that differ fundamentally from test LPD". However example in appendix seems to suggest otherwise?

The aim of the discussion of Watanabe's lemma 3 seems to be to "have some theoretical grasp o nhow temperature affects the test LPD...". However, it is a bit unclear to me, what conclusions you draw from this discussion?

"As beta can be used to capture aleatoric uncertainty in the data..." I am not sure exactly what is meant here. For example in the regression setting you outline, it is assumed that the variance is known and fixed. This would be the aleatory uncertainty, right? The temperature then simply scales this variance in the likelihood. Is there an assumption of model mismatch?

In the regression setting, does the maximum likelihood estimate in eq. 7 differ from a standard maximum likelihood estimate with a learned homoschedastic noise?

Is it correctly understood, that your proposed method consists of first estimating the temperature using maximum likelihood and then estimating the posterior distribution of the model parameters using SGMCMC?

In the regression examples for beta=1 how is the model variance chosen?

How useful is lemma 3.1 when it only holds for the tempered model?

Is there a typo on page 14 line 748? Should the T be there in the last equation?

Minor comments:

"singular models singular models" repeated words

---

### Official Review · Reviewer_Fd9D · 2024-11-02

**Soundness:** 1
**Presentation:** 1
**Contribution:** 1
**Rating:** 3
**Confidence:** 3

**Summary:**

The paper proposes a new way to set the posterior temperature in a more efficient way than simple grid search.
The paper does an MLE on both the weights and the temperature for a tempered prediction distribution to come up with the temperature parameter used for the Bayesian model.

**Strengths:**

I do think that an approach like the proposed could be interesting. It might a good way to find temperatures for tempered Bayesian models, but there is still active debate whether tempering Bayesian models is actually necessary or whether it is an artifact of poor optimization or data augmentations.

**Weaknesses:**

- Written in a misleading way. The authors introduce theory for choosing the inverse temperature $\beta$ that is not part of the standard Bayesian literature in Section 3.0. They do not use it later on, though. This feels like a tactic to make the paper look more complex than it is. What they do in the end is find the temperature using a simple MLE. Just like one would usually train a neural network just with a tempered prediction distribution, which they derive intuitively from a dataset-dependent posterior $\tilde{p}$.
- The justification in 3.2 seems to be much better, but is not presented with enough spotlight and is lacking a lot of cases as it only treats the linear regression case and only does so for the *tempered model* PPD, not the real PPD for which we do the tuning.
- Is this grid search really expensive? I think the actually expensive part should be the SG-MCMC, no? I am missing a time comparison. Thus, the main argument of this paper is not experimentally validated.
- In line 183 you write your new posterior $\tilde{p}$ depends on x, but it depends on the xs in $D$ instead.

**Questions:**

- Why did you include the "singular learning theory" result?
- Why did you not include a time comparison with grid search over temp. values?

---

### Official Review · Reviewer_StPN · 2024-11-05

**Soundness:** 2
**Presentation:** 4
**Contribution:** 2
**Rating:** 5
**Confidence:** 3

**Summary:**

In this paper, the authors propose a likelihood based method for optimising the temperature of the posterior predictive of Bayesian Deep Learning models. The authors discuss the commonly observed fact that the temperature of the posterior predictives obtained from BNNs are sub-optimal, requiring post-hoc optimisation of the temperature. The authors argue that grid search is too expensive, and instead propose to different ways of optimising the temperature of two posterior-predictive density formulation: one in which the temperature is introduced directly in the posterior to obtain a tempered weight posterior, and another in which the temperature is included in the posterior predictive density yielding what the authors call a tempered model. The authors propose tempering during training, finding both optimal weights and temperature during training. The authors evaluate these methods and compare them on a number of datasets. Moreover, they provide two different viewpoints on cold posterior effect (CPE), both from the generalised Bayes and Bayesian Deep learning perspective.

**Strengths:**

* The paper reads very well.
* The topic of this paper is highly relevant with increasing focus on uncertainty quantification of neural networks being highly popular.
* The discussion on the cold posterior effect is interesting, and provides additional insights and bridges understanding between two communities that often are disjoint.

**Weaknesses:**

I will elaborate on these weaknesses in questions.

1. Possible lack of novelty.
2. Overclaims in experimental results.
3. Lacking metrics in experimental results.

**Questions:**

Firstly I would like to say that although I find the paper nice to read, I am not entirely sure how novel the presented methods are, and I find the experimental section quite lacking. I will elaborate on that here:

1. **Regarding lack of novelty**: The authors claim that the use of a hold-out dataset and performing gradient based optimisation for temperature scaling is novel, yet to the best of my knowledge, this is also what is done in [1] when temperature scaling is done. The process of doing this _during_ training of the model parameters may be new, but [1] is never discussed and contrasted to. Could the authors perhaps give their viewpoint on the novelty of their methods in contrast to [1]? And as a minimum, I would expect the authors to compare their methods to these previously proposed temperature scaling methods.
2. **Regarding overclaims**: Unless I am misunderstanding something, I think the statement "The PPDs at β ̸= 1generally outperform SGD and the PPD at β = 1, and our method can achieve comparable, if not better, performance than the grid search" in Table 1 caption is incorrect. In fact there are no cases where the proposed methods statistically significantly outperform grid-search. It may perform on par, but for the claims on cost of grid-search to be validated, some metrics on fit times for the proposed methods would have to be shown.
3. **Regarding lacking metrics**: Could the authors possibly include LPD or ECE in Table 1? I find it strange to not show the LPD in this setting, when uncertainty quantification/calibration is part of the motivation for these methods, but these metrics are never shown when contrasting to other methods.

Minor comments:
1. In line 114 and 115, the authors write that BDL focuses on PPD, while GB focuses on robustness of the model including misspecification and calibrated uncertainty. I would generally disagree with this, as the BDL community largely cares about exactly uncertainty quantification and model calibration. Could the authors possibly comment on this, or as a minimum include some references?
2. The authors never cite [1].



[1]: On Calibration of Modern Neural Networks, Guo et. al, 2017

---

### Meta-Review · Area_Chair_BAmw · 2024-12-20

**Metareview:**

This paper presents a method for data-driven tuning of the temperature when tempering the posterior of a Bayesian deep learning model.  This suggests an interesting approach to addressing the so called "Cold Posterior" problem.  I.e. rather than treat this as a nuisance parameter or as a sign of Bayesian impurity, the authors propose to embrace tempering as a part of the model - i.e. parameterize the temperature and learn it from the data.  The reviewers find the work well motivated, well-written, the approach interesting and the paper clear.  However, the reviewers all voted to reject the paper (5, 3, 5, 3).  The reviewers in general found that the experiments weren't comprehensive enough and that the presented experiments didn't really seem to back up the claims of the authors.   There was some concern that there were confusing or misleading statements in the work.  There were also questions regarding novelty, e.g. compared to Guo et al.  Overall, the consensus sentiment seems to be that the paper isn't really ready from an empirical standpoint - i.e. that the experiments don't comprehensively and convincingly demonstrate that the method works.

Given that the consensus of the reviewers is to reject the paper, the recommendation is to reject.  The reviews seem to suggest that the paper could be made much stronger with more comprehensive empirical evaluation, clearer discussion of how this relates to Bayesian theory and moving metrics like LPD and ECE into the main paper.  Hopefully, these reviews will be helpful in making the paper stronger for a future submission.

**Additional Comments On Reviewer Discussion:**

The reviewers brought up concerns regarding novelty, empirical evaluation, use of certain metrics and a number of requests for clarification.  The authors responded to these concerns in the response.  Only two of the reviewers seemed to really engage in discussion and respond to the authors' response.

The authors seem to have addressed concerns regarding novelty in the rebuttal.  They provided comparisons to Guo et al, that the reviewers seemed to be satisfied with.

One reviewer asked for the ECE and LPD metrics in the experiments.  The reviewer was not satisfied with the authors' response (that there wasn't enough space for these results in the main body of the paper).  The authors seem to have responded to the request for LPD too late - i.e. too late to add it into the main paper and upload it while they could still make changes.  I sympathize with the authors that these results were in the appendix.

The reviewers ultimately weren't satisfied with the response to concerns regarding breadth of empirical evaluation.  This seems like the most significant concern across reviewers and thus weighs highly in my decision.  Ultimately, the reviewers seemed to feel that the claims of the paper were not backed by the presented empirical evidence - i.e. the paper just isn't ready yet for publication.

---

### Decision · Program_Chairs · 2025-01-22

Reject